# Inter-comparison of three AATSR Level 2 (L2) AOD products over China

Y. Che[1,6], Y. Xue[1,2], L. Mei[3], J. Guang[3], L. She[1,6], J. Guo[4], Yincui Hu[5], H. Xu[3], X. He[1,6], A. Di[1,6] and C. Fan[1,6]

[1]State Key Laboratory of Remote Sensing Science, jointly sponsored by the Institute of Remote Sensing and Digital Earth of the Chinese Academy of Sciences and Beijing Normal University, Institute of Remote Sensing and Digital Earth, Chinese Academy of Sciences, Beijing 100101, China

[2]Department of Computing and Mathematics, College of Engineering and Technology, University of Derby, Kedleston Road, Derby DE22 1GB, UK

[3]Key Laboratory of Digital Earth Science, Institute of Remote Sensing and Digital Earth, Chinese Academy of Sciences, Beijing 100094, China

[4]Centre for Atmosphere Watch and Services, Chinese Academy of Meteorological Sciences, 46, Zhongguancun South Avenue, Haidian District, Beijing 100081, China

[5]Hebei Key Laboratory of Environmental Change and Ecological Construction, College of Resources and Environment Science, Hebei Normal University, Shijiazhuang, Hebei Province, China

[6]University of Chinese Academy of Sciences, Beijing 100049, China

*Correspondence to*: Professor Y. Xue (yx9@hotmail.com)

**Abstract.** One of four main focus areas of the PEEX initiative is to establish and sustain long-term, continuous and comprehensive ground-based, airborne and seaborne observation infrastructure together with satellite data. The Advanced Along-Track Scanning Radiometer (AATSR) aboard ENVISAT is used to observe the Earth in dual-view. The AATSR data can be used to retrieve aerosol optical depth (AOD) over both land and ocean, which is an important parameter in the characterization of aerosol properties. In recent years, aerosol retrieval algorithms have been developed both over land and ocean, taking advantage of the features of dual-view, which can help eliminate the contribution of Earth's surface to top of atmosphere (TOA) reflectance. The Aerosol_cci project, as a part of the Climate Change Initiative (CCI), provides users with three AOD retrieval algorithms for AATSR data, including the Swansea algorithm (SU), the ATSR-2ATSR dual view aerosol retrieval algorithm (ADV), and the Oxford-RAL Retrieval of Aerosol and Cloud algorithm (ORAC). The validation team of the Aerosol-CCI project has validated AOD (both Level 2 and Level 3 products) and AE (Level 2 product only) against the AERONET data in a round robin evaluation using the validation tool of the AeroCOM (Aerosol Comparison between Observations and Models) project. For the purpose of evaluating different performances of these three algorithms in calculating AODs over mainland China, we introduce ground-based data from the CARSNET (China Aerosol Remote Sensing Network), which was designed for aerosol observations in China. Because China is vast in territory and has great differences in terms of land surfaces, the combination of the AERONET and CARSNET data can validate the L2 AOD products more comprehensively. The validation results show different performances of these products in 2007, 2008 and 2010. The SU

algorithm performs very well over sites with different surface conditions in mainland China from March to October, but it slightly underestimates AOD over barren or sparsely vegetated surfaces in western China, with mean bias error (MBE) ranging from 0.05 to 0.10. The ADV product has the same precision with a low root mean square error (RMSE) smaller than 0.2 over most sites and the same error distribution as the SU product. The main limits of the ADV algorithm are underestimation and applicability; underestimation is particularly obvious over the sites of Datong, Lanzhou and Urmuchi, where the dominant land cover is grassland, with MBE larger than 0.2, and the main aerosol sources are coal combustion and dust. The ORAC algorithm has the ability to retrieve AOD at different ranges including high AOD (larger than 1.0); however, the stability deceases significantly with increasing AOD, especially when AOD > 1.0. In addition, the ORAC product is consistent with the CARSNET product in winter (December, January and February), whereas other validation results lack matches during winter.

## 1. Introduction

The Pan-Eurasian Experiment (PEEX) is a multidisciplinary, multiscale and multicomponent research, research infrastructure and capacity-building program (Kulmala et al. 2015). One of the strategically most important task of PEEX is to filling the observational gap in atmospheric in-situ data in the Siberian and Far East regions and start the process towards standardized and harmonized data procedures (Kulmala et al. 2011). Aerosols play a major role in Earth's climate system, including intervening in the radiation budget and cloud processes, and affecting air quality and human health (Remer et al., 2005; Samet et al., 2000; Tzanis and Varotsos, 2008; Kokhanovsky and de Leeuw, 2009). The particles suspended in the troposphere scatter solar radiation back to cool the atmosphere or absorb solar radiation, which warms the atmosphere, causing changes in the net effect of aerosols. These particles could also affect the formation and microphysical properties of clouds as cloud condensation nuclei (Andreae and Rosenfeld, 2008). The source of aerosols could be anthropogenic or natural (Varotsos et al. 2012). Particles from different sources are mixed into aerosol masses to influence AOD, reduce visibility (Kinne et al., 2003; Varotsos 2005; Remer et al., 2005) and cause spatial and temporal variability of AOD; therefore, the largest uncertainties in the estimation of radiative forcing are introduced by aerosols (IPCC, 2013).

Over the past 35 years, different types of satellites have been used to obtain atmospheric information, especially aerosol properties (Griggs, 1979; Kokhanovsky and de Leeuw, 2009). Remote sensing provides a means to obtain global and long-term observations of aerosols, especially in the widest ocean and remote regions where ground-based stations cannot be constructed. In addition, polar-orbiting satellites and geostationary satellites obtain daily global images, which helps to capture changes in aerosol patterns and properties (Prins et al., 1998; Torres et al., 2002). There are, however, many difficulties in observing aerosols by satellites because depending on the surface properties, the contribution to the signal received by the satellite can vary drastically; aerosol components and concentrations are constantly varying, and their sources cannot be precisely determined (Levy et al., 2007).

The Advanced Along-Track Scanning Radiometer (AATSR) aboard ENVISAT is used to observe the Earth in dual-view, of which one is nadir direction and the other is forward direction with a viewing angle of 55° from nadir view. The AATSR was designed to have seven spectral channels at wavelength of 0.55, 0.67, 0.87, 1.63, 10.7 and 12 μm. The nadir spatial resolution is 1 km × 1 km with a swath width of 512 pixels. Furthermore, the AATSR instrument equipped two calibration targets, black-body calibration target for thermal channels and opal visible calibration target for visible and near-Infrared channels, aiming to implement self-calibration. The data from AATSR can be used to retrieve AOD both over land and ocean, which is important for the characterization of aerosol properties (Adhikary et al., 2008). In recent years, some aerosol retrieval algorithms have been established both over land and ocean, taking advantage of the features of dual-view, which can help eliminate the contribution of surface to top of atmosphere (TOA) reflectance. Aerosol_CCI, as part of the Climate Change Initiative (CCI) (http://www.esa-aerosol-cci.org/), provides users with three algorithms for AATSR data, including the Swansea algorithm (SU) (Bevan et al. 2012), the ATSR-2/AATSR dual-view aerosol retrieval algorithm (ADV) (Kolmonen et al. 2015) and the Oxford-RAL Retrieval of Aerosol and Cloud algorithm (ORAC) (Thomas et al. 2009). The aim of this work is to evaluate different performances of these algorithms in calculating AOD over different regions of China in 2007, 2008 and 2010.

A ground-based sun−photometer has been used to take sun and sky measurements directly (Holben et al., 1998). The Aerosol Robotic NETwork (AERONET) has constructed hundreds of sites all over the world as of 2015. These stations, operated by the American National Aeronautics and Space Administration (NASA), are operational worldwide, providing multi-spectral channel validation data for satellite-retrieved data to complete synthetic measurements on a global scale.

The China Aerosol Remote Sensing Network (CARSNET) is a ground-based aerosol monitoring system that uses CE-318 sun-photometers, similar to AERONET, and has constructed 37 sites throughout China (Che et al., 2009). It has been validated that CARSNET AOD measurements are approximately 0.03, 0.01, 0.01 and 0.01 larger than measurements of AERONET at the 1020, 870, 670 and 440 nm channels, respectively (Che et al., 2009). In this paper, we combine two aerosol observation datasets from AERONET and CARSNET as reference data to validate these three AATSR AOD products over China more comprehensively.

The basic method for assessment is to compare the retrieval results with data (AOD mainly) obtained by AERONET/CARSNET. However, this direct comparison of retrieval results with AERONET data is limited due to different cloud screening processes (de Leeuw et al., 2013), and such a limitation could influence the validation reliability to some extent. To make the validation more reliable, comparison of the retrieval results with high quality data from MODIS or MISR is also one effective method for validation (Kahn et al., 2009). However, AERONET or other ground-based networks provide accurate measurements without the influence of land surface reflection (Holben et al., 1998), which means that comparison of retrieved AOD with ground-based measurements is the basic method. The AATSR L2 products provided by Aerosol_CCI

have been validated by the validation team via a round robin (RR) test (de Leeuw et al., 2013). On this basis, we focused on assessing the performances of AATSR aerosol L2 products in mainland China by comparing the retrieval results with AERONET and CARSNET data.

## 2. Reference data and validation statistics

AOD is the most important parameter in terms of aerosol properties and is different from other retrieved parameters under the project of Aerosol_CCI. The Aerosol_CCI project adopts three aerosol retrieval algorithms for ATSR-2/AATSR instrument, including Swansea algorithm (SU) (Bevan et al. 2012), the ATSR-2/AATSR dual view aerosol retrieval algorithm (ADV) (Kolmonen et al. 2015) and the Oxford-RAL Retrieval of Aerosol and Cloud algorithm (ORAC) (Thomas et al. 2009b). All of these three algorithms have ability in retrieval of aerosol properties both over land and ocean. ADV algorithm was originally

developed for retrieving AOD properties over land at wavelength of 0.555, 0.659 and 1.61 μm (Veefkind et al. 1998). The main advantage of ADV is the introduction of k-ratio approach to eliminate contribution of reflection to TOA reflectance, which uses the ratio of the reflectance measured in the forward and nadir views (Flowerdew and Haigh, 1995). The ORAC algorithm is designed to retrieve AOD properties at each of four AATSR short-waves channels both over land and ocean, including AOD, effective radius and surface reflectance. The build of the forward model used in ORAC algorithm is based on

radiative transfer code - DISORT. A parameterized model of surface reflectance distribution is used in retrieval and combines with the AATSR dual-view to make up shortage of the need of a priori of reflectance (North et al. 1999). An iterative optimization method is employed to determinate AOD, aerosol type and surface reflectance.

AATSR L2 data (see Tab. 1) are daily products with a spatial resolution of $10 \times 10 \ km^2$, and contain a quality flag or a level of confidence for each pixel (de Leeuw et al., 2013). Compared to the Level 3 (L3) product with a spatial resolution of $1° \times 1°$,

daily L2 data have higher spatial resolution, which helps to capture greater detail of aerosol properties and is further explored in our follow-up study.

**Tab. 1. Details of AATSR AOD products.**

| algorithm | version | sensor | Main parameters | Resolution coverage |
|-----------|---------|--------|-----------------|---------------------|
| ADV/ASV | 2.3 | AATSR | AOD,ANG | 10 km, 1° global |
| SU | 4.21 | AATSR | AOD,ANG | 10 km, 1° global |
| ORAC | 03.04 | AATSR | AOD, aerosol type | 10 km, 1° global |

It has been demonstrated that the ground-based observation data from the AERONET have the ability and precision to be used as reference data when users validate AOD (Holben et al., 1998). There are eight AERONET sites in mainland China providing

Level 2.0 (L2) data (cloud-screened and quality-assured) for 2007, 11 sites for 2008 and 10 sites for 2010, from which the AOD measurement data are available on the website. However, most of these sites are distributed in the eastern China coastal area, as shown in Fig. 1, which, however, does not meet the requirements of comprehensively validating the aerosol properties over all of China. Substantial hazardous aerosol pollution affects most regions of northern (Li, 2014) and eastern China in winter, and heavy dust aerosols from the Taklimakan desert in western China can be transported long distances to eastern China, even to Japan (Takahashi, 2011), resulting in regional differences.

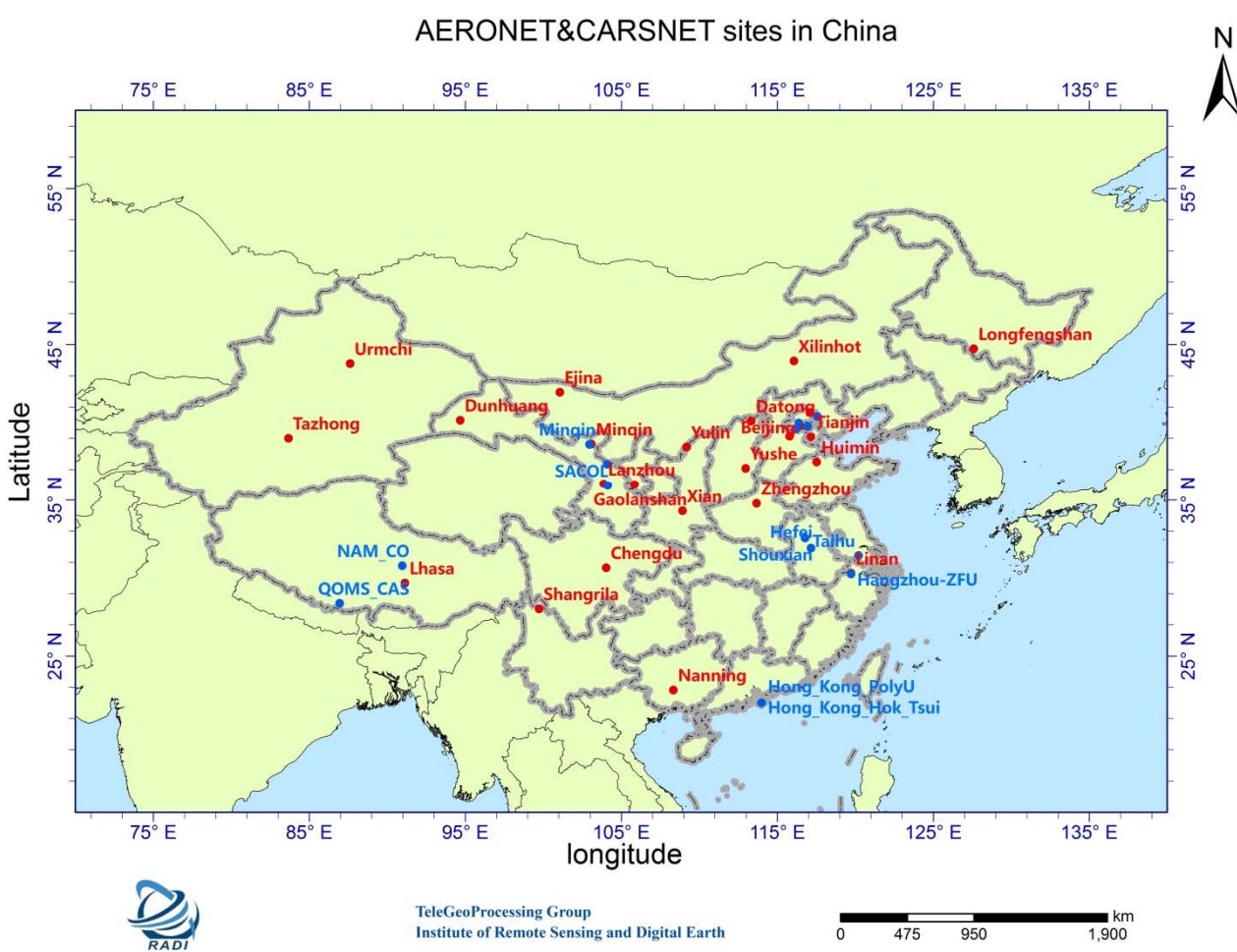

**Fig. 1. The distribution of selected AERONET&CARSNET sites in mainland China in 2007, 2008 and 2010. The blue and red points represent AERONET and CARNET sites, respectively.**

The measurements from another network, the CARSNET, equipped with calibrated CE-318 instruments, have the same accuracy as AERONET. The CARSNET has more sites than the AERONET in mainland China, and the spatial distribution of the CARSNET sites is distributed more evenly. Therefore, for the purpose of assessing different performances of these three AATSR L2 AOD products, we selected ground-based measurements from both of these two networks as reference data.

The AERONET provides AOD data at three data quality levels: Level 1.0 (unscreened), Level 1.5 (cloud-screened), and Level 2.0 (L2) (cloud-screened and quality-assured) (http://aeronet.gsfc.nasa.gov/new_web/index.html). Here, we selected AERONET L2 data that are screened and quality-assured. Because both the AERONET and CARSNET data are AATSR products without band-effective wavelengths, we interpolated the ground-based data to the 550-nm wavelength. The AOD of the L2 datasets were compared with AERONET&CARSNET observation data using scatter plots and linear-regression of the

data. The comparisons were made for collocated satellite and ground-based observations (Ichoku et al., 2002), i.e., AOD pixels were selected within a spatial extent of $\pm25$ km of ground-based stations and a time range of $\pm30$ min of the AATSR overpass from the ground-based measurements. At least five AATSR AOD retrievals and two AERONET/CARSNET observations are required in each collocation (Levy et al., 2010).

We conducted collocations according to year (2007, 2008 and 2010) and dataset (ADV, ORAC and SU). In total, 20 ground-

15 based observation sites, including 12 AERONET sites and 8 CARSNET sites, were in the Chinese territory in 2007, of which six AERONET and eight CARSNET inland sites were selected. For 2008, we selected 8 AERONET and 24 CARSNET inland sites, for a total of 32 sites, ignoring the island sites and those near the shoreline. For 2010, only six CARSNET sites are available for us, and a total of 14 inland sites were selected with eight AERONET inland sites (see Table 2).

**Table 2. Selected ground-based sites in China.**

|      | Network | inland | near shoreline | island | Total |
|------|---------|--------|----------------|--------|-------|
| 2007 | AERONET | 6      | 6              | 0      | 12    |
|      | CARSNET | 8      | 0              | 0      | 8     |
|      | Total   | 14     | 6              | 0      | 20    |
| 2008 | AERONET | 8      | 7              | 0      | 15    |
|      | CARSNET | 24     | 1              | 0      | 25    |
|      | Total   | 32     | 8              | 0      | 40    |
| 2010 | AERONET | 8      | 7              | 1      | 16    |
|      | CARSNET | 6      | 0              | 0      | 6     |
|      | Total   | 14     | 7              | 1      | 20    |

**2.1 Statistical Metrics**

Collocated pairs are analysed using statistical methods. Bias describes the average difference between satellite retrievals and ground AOD. Then, to determine how well the satellite data match the ground-based observation data, the relationship between them is explored. Some basic statistics are shown on the scatter plot, including the root mean square error (RMSE):

$$\text{RMSE} = \sqrt{\frac{1}{n}\sum_{i=1}^{n}\left(\tau_{sat,i} - \tau_{aero,i}\right)^2} \tag{1}$$

where $\tau_{aero,i}$ represents the ground-based observation data and $\tau_{sat}$ represents the satellite retrievals. Mean satellite-retrieved AOD (MSA) and mean AERONET&CARSNET AOD (MAA) represent the central tendency of the data. Relative mean bias (RMB) is used to determine under- or overestimation of the AOD retrievals; it is the ratio of MSA to MAA:

$$\text{RMB} = \text{MSA/MAA} \tag{2}$$

Mean bias error (MBE) is the mean difference between the satellite retrievals and AATSR AODs, and the mean absolute error (MAE) is the absolute value of the mean bias error. Together with RMB, the MBE and MAE are used to determine the
magnitude of the difference between the two datasets.

**2.2 KAPPA Statistics**

In the scatter plot of the collocated pairs, the retrieved data and the corresponding collocated ground-based observation data could be considered as two arrays, and the main purpose of KAPPA is to explore how these two arrays match each other. For retrieval of aerosol properties, the performances of most algorithms decrease in effectiveness with increasing AOD, i.e.,
difficulties in retrieving AOD will be increased as AOD increases. Obviously, when only using $|bias|$, the absolute value of the difference between ground-based data and AATSR AOD data in each collocation pair, as an assessment standard for different AODs, is insufficient and lacks persuasion. Therefore, the combination of $|bias|$ and $|bias|/Ground$, i.e., the ratio of $|bias|$ to the value of the reference data in each collocation pair, used in the KAPPA coefficient will account for this shortage and provide a new statistic for assessing the agreement between two arrays, taking advantage of the KAPPA coefficient.

The KAPPA coefficient was originally proposed as a descriptive statistic indicating the degree of beyond-chance agreement between two ratings per subject in a dichotomous form (Bloch and Kraaemer, 1989). KAPPA coefficients with various forms also could be used to measure the accuracy of thematic classifications (Rosenfield and Fitzpatrick-Lins, 1986). KAPPA is, in short, a measure of "true" agreement (Cohen, 1960). The pairs collocated by matching ground-based data with AATSR L2

AOD data could be regarded as two different arrays so that we introduced the KAPPA coefficient to assess agreement between these two arrays. Based on the concept of the KAPPA coefficient proposed by Cohen (1960), an appropriate modification with a two-category nominal scale is shown in Table 3.

**Table 3. Design of the KAPPA coefficient.**

| | | Criterion 2 | | Total |
|---|---|---|---|---|
| | | Relevant (highly) | Relevant (low) | |
| Criterion 1 | Relevant (highly) | a | b | G1 |
| | Relevant (low) | c | d | G2 |
| Total | | F1 | F2 | n |

To estimate the KAPPA coefficient, one needs to determine which pairs are "true" or which pairs are "relevant". However, if only given matched collocation pairs, we cannot determine which pair is relevant. Therefore, the design of criterion 1 and criterion 2 needs to be reasonable and fit for the purpose of validation.

For criterion 1, if $|bias|$ is greater than the mean of $|bias|$, then it is marked as "far from truth", and if not, it is marked as "close to truth". Here, the bias was assessed from the first quartile to the third quartile for eliminating possible "outliers". The

$|bias|$ only indicates the absolute error of the retrieved AOD, and it still needs another statistic for criterion 2, i.e., $|bias|/Ground$, which indicates the relative error of AOD retrieval. For criterion 2, if $|bias|/Ground|$ is greater than 0.2 (according to EE4), then it is marked as "far from truth", and if not, it is marked as "close to truth". For the conventional formula of calculating the KAPPA coefficient:

$$K = \frac{P_0 - P_c}{1 - P_c} \tag{3}$$

where $P_o$ is the proportion of observed agreement and $P$ is the proportion of chance agreement.

$$P_0 = \frac{(a + d)}{n} \tag{4}$$

$$P_c = \frac{\left(\frac{F_1 \times G_1}{n}\right) + \left(\frac{F_2 \times G_2}{n}\right)}{n} \tag{5}$$

Algorithms for AATSR AOD retrieval used to underestimate AOD over different regions in China include the ADV ORAC and SU algorithms. On this basis, the agreement between ground-based observation data and satellite retrievals is assessed

based on the ADV and SU algorithms (Che et al., 2015). The main aim of this new KAPPA coefficient is to evaluate the comprehensive performance of these algorithms. Its function is to represent not only the degree of underestimation but also the level of agreement between different datasets.

## 3. Validation results and analysis

5   We collected different validation reference data of AERONET and CASNET in 2007, 2008 and 2010. Only 14 ground-based observation sites are available in 2007, of which some are located close to each other. Most are located in different provinces; however, the total number of sites is small and the space distribution is not uniform. Therefore, the number of matches is relatively small for all of the algorithms. More AERONET/CARSNET data are available in 2008, with a total of 32 sites including 8 AERONET sites and 24 CARSNET sites. There are 14 AERO&CARS sites providing data for validation in

10   2010.The focus of this paper is to determine the differences between the ADV, ORAC, and SU L2 AOD products (see Tab. 4 and Figure 1). Therefore, we calculated statistics and analysed the validation results separately by year (see Tab. 4 and Figure 1).

**Table 4. Main statistics of the validation results.**

|  |  | N | MSA | MAA | MBE | MAE | RMSE | RMB | KAPPA |
|---|---|---|---|---|---|---|---|---|---|
| AATSR ADV | 2007 | 94 | 0.25 | 0.36 | -0.11 | 0.12 | 0.15 | 0.70 | 0.49 |
|  | 2008 | 327 | 0.22 | 0.36 | -0.14 | 0.15 | 0.20 | 0.61 | 0.37 |
|  | 2010 | 147 | 0.17 | 0.31 | -0.14 | 0.14 | 0.22 | 0.55 | 0.23 |
|  | 3Years | 568 | 0.21 | 0.35 | -0.13 | 0.14 | 0.20 | 0.61 | 0.38 |
| AATSR ORAC | 2007 | 145 | 0.35 | 0.28 | 0. 06 | 0.14 | 0.23 | 1.23 | 0.50 |
|  | 2008 | 648 | 0.29 | 0.33 | -0.04 | 0.16 | 0.27 | 0.87 | 0.45 |
|  | 2010 | 298 | 0.26 | 0.27 | -0.01 | 0.14 | 0.23 | 0.96 | 0.37 |
|  | 3Years | 1091 | 0.29 | 0.31 | 0.02 | 0.15 | 0.25 | 0.93 | 0.44 |
| AATSR SU | 2007 | 98 | 0.33 | 0.41 | -0.07 | 0.09 | 0.16 | 0.83 | 0.43 |
|  | 2008 | 446 | 0.29 | 0.41 | -0.12 | 0.13 | 0.21 | 0.72 | 0.50 |
|  | 2010 | 171 | 0.27 | 0.37 | -0.10 | 0.12 | 0.21 | 0.73 | 0.53 |
|  | 3Years | 715 | 0.29 | 0.40 | -0.11 | 0.12 | 0.20 | 0.73 | 0.50 |

### 3.1 Validation results

### 3.1.1 The ADV algorithm

For 2007, the RMS error is 0.095 and the RMB is 0.704, which reflects the tendency of underestimation. This type of underestimation is more severe with increasing AOD. Low dispersion and slight underestimation make the KAPPA coefficient
high (0.473), demonstrating that the ADV algorithm performs well in calculating the AOD over China in 2007. The ADV algorithm is appropriate for the retrieval of low AODs, especially for those less than 1.0; thus, the MSA for 2007 is 0.244. For 2008, the lower RMB (0.621) suggests more severe underestimation, and higher RSE (0.130) indicate lower accuracy. Similar with 2007, the MSA of the ADV is 0.211. Therefore, the KAAPA coefficient, which measures the overall performance, is 0.329, lower than that of 2007. For 2010, the lowest RMS (0.089) with the lowest accidental error of the three years. However,
the KAPPA coefficient is 0.180, also the lowest of the three years. The most obvious feature of the ADV algorithm is underestimation with the highest MSA is 0.250 in 2007, and the lowest is 0.173 in 2010. The ADV algorithm can retrieve low AOD values with high accuracy. This "ability" is systematic for either high AODs or low AODs. This also limits the range of application of the ADV algorithm, especially in calculating AODs in high value ranges.

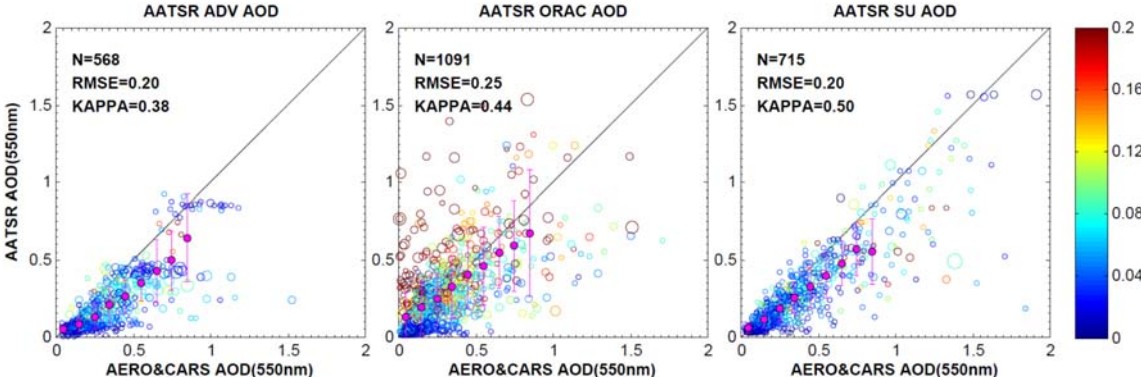

**Fig. 1 Scatter plots of AATSR ADV, ORAC and SU L2 AOD products with ground-based data in China for three year of 2007, 2008 and 2010. The black solid line represents 1-1 line. The magenta points are means for specific range of AERO&CARS AOD and the magenta lines are mean $\pm 2\sigma$ of retrievals at certain range. The areas and colours are determined by means of uncertainty (MU) dataset in AATSR L2 products and standard deviation of retrievals (Std_S) in collocation frame of 50 km $\times$ 50 km respectively.**

### 3.1.2 The ORAC algorithm

The ORAC algorithm performed well for 2007, achieving a KAPPA coefficient of 0.474. However, the distribution of matches is dispersed implying high RMSE (0.206). In terms of the degree of fitness, its performance is not effective. However, there is no obvious trend of underestimation or overestimation, and accidental errors influence the accuracy of the ORAC algorithm. The MSA of the ORAC is 0.324. ORAC has the most matches of the three algorithms. Different from 2008, no obvious underestimation occurs in the results of 2007 and 2010. For 2008, the RMB is 0.829, suggesting a slight underestimation trend. The applicability of ORAC is high, with MSA of 0.271. The collocated pairs are relatively dispersed, influencing the RMSE. For 2010, the same dispersion of points in the scatter plot and low KAPPA coefficient are observed. Overall, the ORAC algorithm tends to retrieve AODs unstably for either high AODs or low AODs and with slight underestimation in 2007. The results of 2008 and 2010 share common features, indicating that accidental error is larger than systematic error.

### 3.1.3 The SU algorithm

The SU algorithm performed well for all three years, achieving KAPPA coefficients of 0.409, 0.484 and 0.520, respectively. The RMBs are 0.816, 0.713 and 0.720 for 2007, 2008 and 2010, respectively, demonstrating the underestimation of the SU product. The applicability of SU is high, with MSA of 0.293 for 2008. The most obvious feature of the SU algorithm is its stability in retrieving AOD for different years or different regions (Fig. 4). The MSA ranges from 0.270 for 2010 to 0.330 for 2007, and the KAPPA coefficient is from 0.520 to 0.409, which suggests that the SU algorithm performed better in retrieving low AODs. The SU algorithm has the best performance in terms of AOD retrieval, as it has the highest KAPPA coefficient (0.520). Overall, the SU algorithm can be applied to retrieve AOD in different ranges with high precision. Factors influencing the performance of the SU algorithm include small systematic error and even smaller accidental error.

### 3.2 Uncertainty analysis based on aerosol loading

In the previous section, we validated all three AOD products over mainland China in 2007, 2008 and 2010, discovering that all three products tend to exhibit underestimation to some extent. For the purpose of ascertaining the causes of the underestimation, in this section, we focus on analysing the AOD uncertainties leading to differences between retrieved AODs and ground-based AODs in special conditions. Collocated pairs are divided into three groups according to aerosol loading, including light loading ($\tau < 0.15$), heavy loading ($\tau > 0,4$), and moderate loading (Levy et al., 2010). It is obvious that the AOD bias increases with increasing AOD for all three products. These products have one feature in common, that is, the AOD bias tends to be negative, which indicates that the underestimation becomes more significant with increasing aerosol loading. The ADV and SU algorithms perform well in estimating AOD, i.e., with little underestimation (lower MBEs of -0.04 and -0.02 respectively as shown in Tab. 5), when aerosol loading is low (light loading) (Fig. 2).

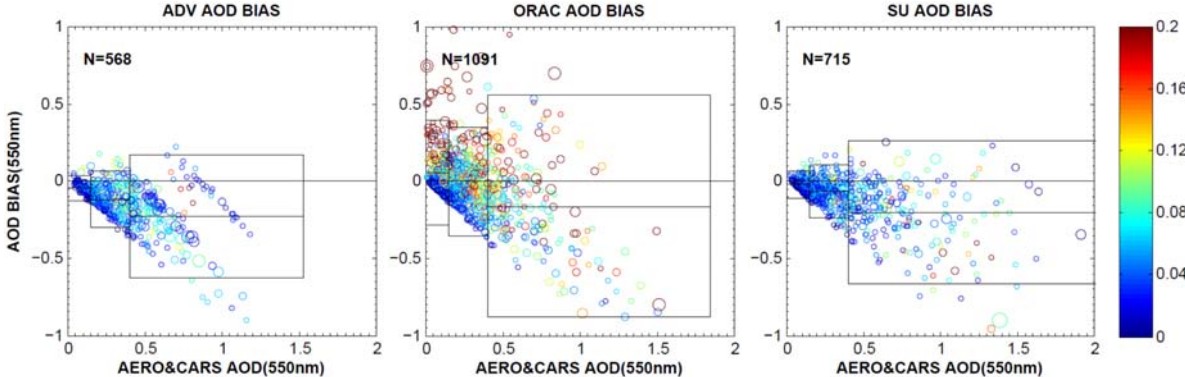

**Fig. 2. Scatter plot of AERONET&CARSNET AODs with ADV AOD bias or uncertainties in China in 2007, 2008 and 2010. The areas and colours of bubbles represent MU and Std_S sampling area of 50km × 50km respectively. Colours represent different groups: blue denotes light loading, green denotes moderate loading, and red denotes heavy loading. Each group has one box, the bottom and top borders of which represent MBE + 2σ and MBE − 2σ, respectively, containing 96% of scattered points from each group. The centre line of each box represents the MBE of each group.**

Under complex conditions, the ORAC overestimates AOD in regions of light loading and moderate loading compared with the AERONET and CARSNET, as shown in Fig. 2. ADV tends to underestimate AOD more severe with MBE = -0.11 in moderate aerosol loading region. Similar with ADV, the underestimation of SU in moderate aerosol loading becomes more severe with MBE = -0.07. ORAC performs the best in retrieving in moderate aerosol loading region without underestimation or overestimation, even though the bubbles distribute discrete with Std of 0.18 (see in Tab. 5). The performances of all three algorithms are at same level with close MBEs, Stds and RMSEs in heavy aerosol loading region.

The top and bottom borders of the box we draw represent the interval of $[-2\sigma, 2\sigma]$, which contains most of the data (approximately 95%) for a given group. The data outside the box are "possible outliers" based on the largest error contained in each group. Those "possible outliers" have one feature in common in that the corresponding points in the bias scatter plot are far away from other points. Otherwise, the points below or above the box are different. If a point is above the box, which indicates that the satellite-retrieved AOD is larger than the ground-based observed AOD, this "outlier" tends to be caused by a residual cloud. The ground-based network measures AOD from only one point; however, the satellite-retrieved AODs in each collocated pairs are an average of 25 pixels. Any one of these 25 pixels with a cloud residual will lead to an increased AOD in a collocated pair. Therefore, we conclude that the "outliers" above the box are possibly caused by a cloud residual. From this view, there are 6, 6 and 2 bubbles above the each box for the ADV product for light, moderate and heavy aerosol loading respectively. However, these bubbles are not "possible outlier" due to the MUs and Std_Ss are relative small shown in Fig. 2. Similarly, the bubbles from SU product above each box are not "possible outliers". For the ORAC product, most of bubbles above each box are "possible outliers" due to larger Std_S (>0.2). Most of "possible outliers" are concentrated in light

(13 bubbles) and moderate (14 bubbles) aerosol loading regions as shown in Tab. 5, influencing ORAC's performance on estimating AOD. The bubbles below the box are different from those above the box. Most of them are only below the boxes of moderate and heavy aerosol loading, indicating that all these algorithms have limitations of underestimation on estimating AOD in moderate and heavy aerosol loading regions, especially when AOD loading increase.

We make these groups because aerosols exhibit different behaviours with different loading conditions. In general, the bias or uncertainty of satellite-retrieved AOD will increase with increasing AOD or aerosol loading. As discussed above, all of these algorithms underestimate AOD at different levels; similarly, it is worth noting that underestimation becomes more severe with increasing AOD or aerosol loading.

**Table 5. Statistics of comparison between AOD bias and ground-based measurements. Proportion is the ratio of numbers of bubbles**
**falling in each box to total. RMSE1 and RMSE2 are RMSEs of AOD bias with ground-based measurement and AOD uncertainty respectively.**

| Algorithm | Class | N | Proportion | MBE | RMSE_ | Std | Above | Below | RMSE2 |
|---|---|---|---|---|---|---|---|---|---|
| AATSR ADV | Total | 568 | 100.0% | -0.13 | 0.20 | 0.15 | 14 | 14 | 0.27 |
| | Light | 126 | 22.2% | -0.04 | 0.03 | 0.04 | 6 | 0 | 0.07 |
| | Moderate | 259 | 45.6% | -0.11 | 0.10 | 0.09 | 6 | 6 | 0.22 |
| | Heavy | 183 | 32.2% | -0.23 | 0.17 | 0.20 | 2 | 8 | 0.35 |
| AATSR ORAC | Total | 1091 | 100.0% | 0.02 | 0.25 | 0.25 | 32 | 8 | 3.65 |
| | Light | 347 | 31.8% | 0.059 | 0.10 | 0.17 | 13 | 0 | 0.13 |
| | Moderate | 468 | 42.9% | 0 | 0.11 | 0.18 | 14 | 2 | 0.28 |
| | Heavy | 276 | 25.3% | -0.16 | 0.20 | 0.36 | 5 | 6 | 5.56 |
| AATSR SU | Total | 715 | 100.0% | -0.11 | 0.20 | 0.17 | 8 | 22 | 0.24 |
| | Light | 147 | 20.6% | -0.02 | 0.02 | 0.04 | 3 | 1 | 0.08 |
| | Moderate | 306 | 42.8% | -0.07 | 0.07 | 0.08 | 5 | 12 | 0.16 |
| | Heavy | 262 | 36.6% | -0.2 | 0.19 | 0.23 | 0 | 9 | 0.34 |

Additionally, we make comparison of AOD bias, which is retrieval errors observed, with AOD uncertainty in AOD retrieval for each pixel from AATSR L2 dataset. AOD retrieval error observed (AOD bias) and AOD uncertainty in retrieval are different as evaluating merits. The range of SU AOD uncertainty is from 0.025 to 0.3, smaller than others, even in heavy
aerosol loading region. Most of bubbles of ADV product in Fig.3. are from 0 to 0.4 of AOD uncertainty. The AOD bias and uncertainty are small in light aerosol loading and moderate for ADV and SU products as shown in Fig. 3. For ORAC product, there is no obvious regularity between AOD bias with AOD uncertainty in three aerosol loading regions especially those bubbles with high Std_S.

3.3 Uncertainty analysis of individual ground measurement sites

For the purpose of further evaluating the different performances of these three algorithms in estimating AOD over mainland China, we validate these products on a site-by-site basis. It is significant to explore the roles of different factors in estimating AOD. There are several factors that may have impacts on AOD calculation, including land cover, aerosol type, elevation, etc.

Therefore, we analyse different validation results of each site to study how these factors work (see Table 6).

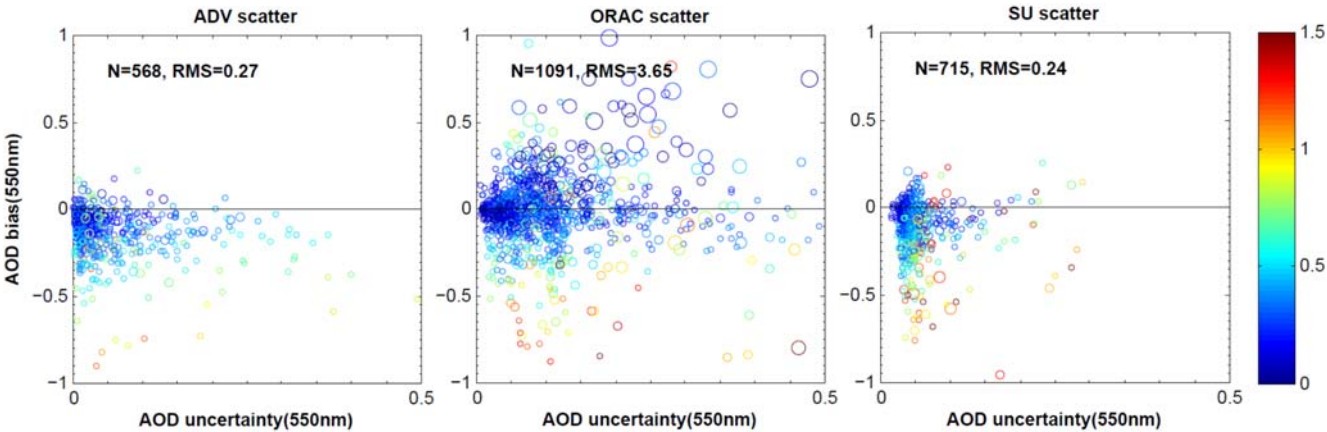

**Fig. 3. Scatter plots of ADV, ORAC and SU AOD uncertainty with AOD bias over China for three years of 2007, 2008 and 2010. The area and colours of bubbles represent Std_S and AOD respectively.**

### 3.3.1 Inter-comparison of algorithms site by site

In this section, we select five representative AERONET&CARSNET sites with more than 30 successful matches in 2007, 2008 and 2010 to guarantee an appropriate statistical sample size. These selected sites are located in different regions where the land cover and climatic pattern are different and representative of mainland China. Two AERONET sites and three CARSNET sites were selected, including SACOL and XiangHe from AERONET, and Linan, Shangdianzi and Xilinhot from CARSNET. Most matches of ADV and SU products collocated with ground-based data occurred in March to October in 2007,

2008 and 2010, as shown in Fig. 4 to Fig. 8. The matches of the ORAC product were distributed in each month over most sites.

Linan is located at 30.3°N, 119.73°E, northwest of Zhejiang province. A total of 80% of the 50 km × 50 km surrounding area is covered by green vegetation, and the other 20% is covered with urban land. The ADV and ORAC algorithm underestimated AOD, with MBE = 0.13 and 0.12 in 2010, respectively. The SU performed well in Linan, with slight underestimation. The

underestimation of the ADV algorithm is more severe than that of SU and ORAC. Although the ORAC algorithm has the most matches in Linan, its performance was unstable, which means that the level of underestimation was different in different years.

**Table 6. Statistics of validation results of different products over different sites.**

| Site | Algorithm | N | MSA | MAA | MBE | MAE | RMSE | RMB | KAPPA |
|------|-----------|---|-----|-----|-----|-----|------|-----|-------|
| Linan | ADV | 33 | 0.346 | 0.462 | -0.116 | 0.122 | 0.088 | 0.748 | 0.341 |
| | ORAC | 48 | 0.426 | 0.470 | -0.044 | 0.131 | 0.144 | 0.906 | 0.668 |
| | SU | 40 | 0.430 | 0.484 | -0.054 | 0.082 | 0.093 | 0.889 | 0.650 |
| SACOL | ADV | 46 | 0.156 | 0.285 | -0.129 | 0.132 | 0.068 | 0.547 | 0.283 |
| | ORAC | 74 | 0.286 | 0.314 | -0.028 | 0.102 | 0.170 | 0.910 | 0.595 |
| | SU | 49 | 0.265 | 0.291 | -0.027 | 0.062 | 0.072 | 0.908 | 0.878 |
| Shangdianzi | ADV | 52 | 0.172 | 0.297 | -0.125 | 0.131 | 0.087 | 0.578 | 0.339 |
| | ORAC | 66 | 0.267 | 0.304 | -0.037 | 0.107 | 0.134 | 0.879 | 0.407 |
| | SU | 46 | 0.285 | 0.402 | -0.117 | 0.128 | 0.101 | 0.710 | 0.457 |
| XiangHe | ADV | 33 | 0.184 | 0.284 | -0.100 | 0.102 | 0.070 | 0.649 | 0.169 |
| | ORAC | 34 | 0.227 | 0.240 | -0.013 | 0.091 | 0.096 | 0.946 | 0.577 |
| | SU | 36 | 0.368 | 0.392 | -0.024 | 0.058 | 0.077 | 0.939 | 0.444 |
| Xilinhot | ADV | 49 | 0.082 | 0.198 | -0.116 | 0.117 | 0.046 | 0.414 | 0.148 |
| | ORAC | 110 | 0.190 | 0.182 | 0.008 | 0.109 | 0.166 | 1.043 | 0.389 |
| | SU | 61 | 0.140 | 0.220 | -0.081 | 0.085 | 0.063 | 0.634 | 0.444 |

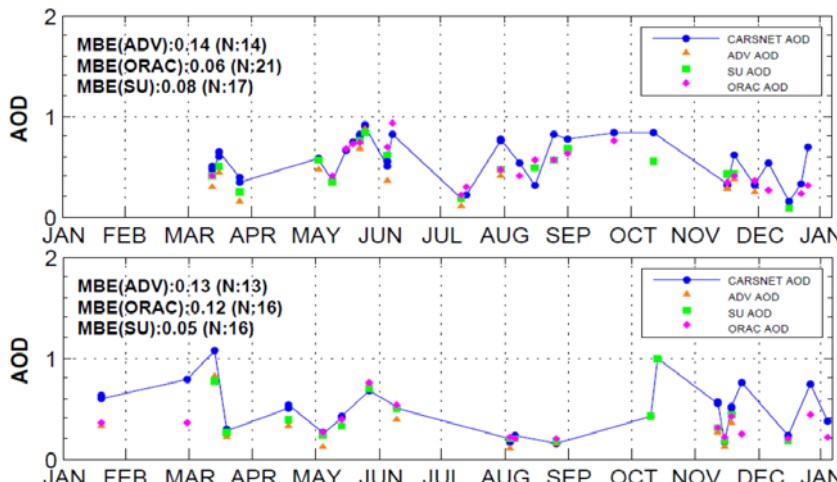

**Fig. 4. Time series comparison of AATSR AOD with CARSNET AOD at Linan in 2008 and 2010.**

SACOL is situated along the southern bank of the Yellow River in Lanzhou city, Gansu province. Lanzhou city has a temperate continental climate with four clearly distinctive seasons. The dominant land cover is grassland, covering approximately 95% of the spatial extent of the 50 km × 50 km area from the MODIS MCD12C1 land cover data. A total of 30% of the surface is arid and semi-arid areas, which can be a source of dust aerosols. SU performs well in retrieving AOD over SACOL, with a low RMSE (0.072). The accidental error in the retrievals using the ORAC algorithm is obvious, leading to a high RMSE (0.170). However, as discussed above, the ADV algorithm severely underestimated AOD in SACOL. The ADV algorithm tended to severely underestimate the AOD of different ranges, except for a small number of high quality matches. The matches of the SU product are of high quality for the three years. The ORAC has collocated matches in January, February, November and December (winter time), unlike the ADV and SU products. However, the accuracy of ORAC in winter is highly uncertain, as shown in Fig. 5.

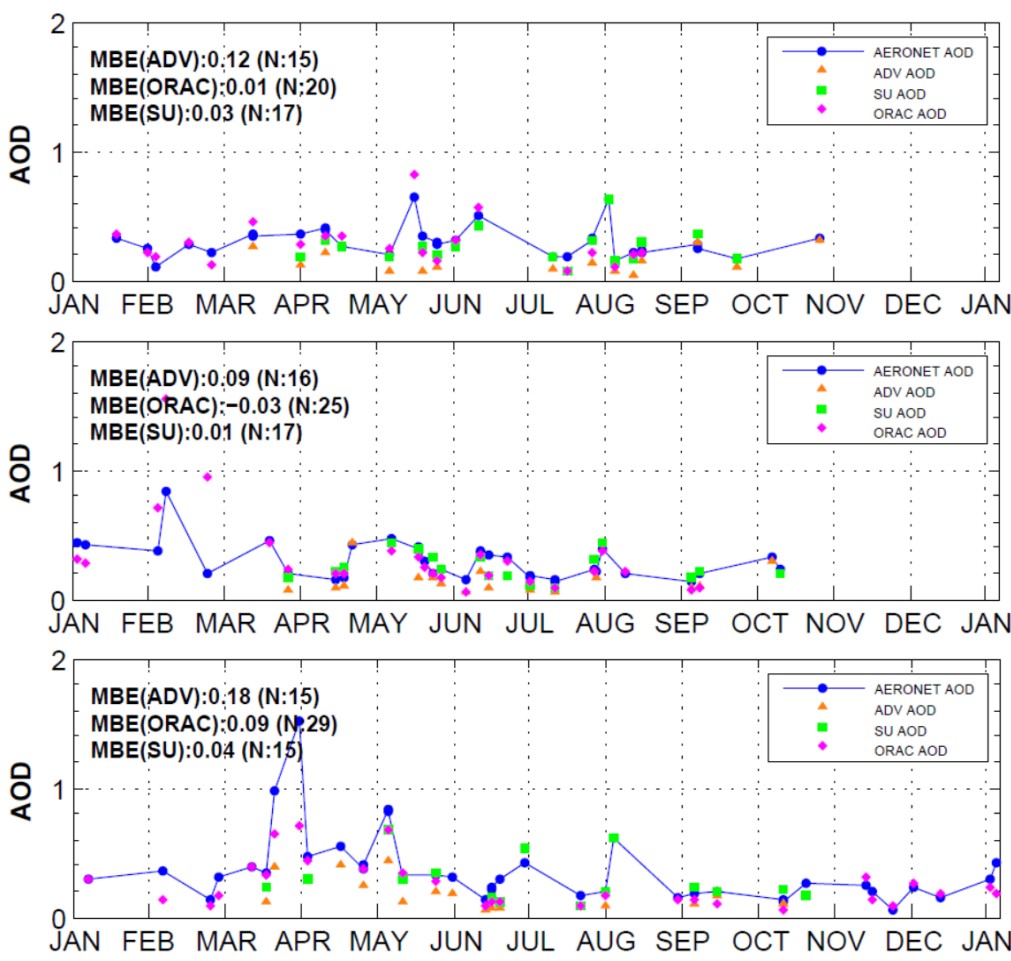

Fig. 5. Time series comparison of AATSR AOD with AERONET AOD at SACOL in 2007, 2008 and 2010.

Shangdianzi is situated at 40.15°N, 94.68°E, with complex land cover of approximately 45% cropland, 30% mixed forest, 18% closed shrub land, 5% grassland, 1% water and 1% evergreen needle leaf forest. The SU algorithm has high precision of AOD calculation over this site from March to October, when most of the land cover is green. The ADV algorithm also performs well in calculating AOD over these three sites, with slight underestimation. The performance of the ORAC algorithm in Shangdianzi is unstable, with strong agreement with ground-based data from March to October and severe underestimation in winter, as shown in Fig. 6.

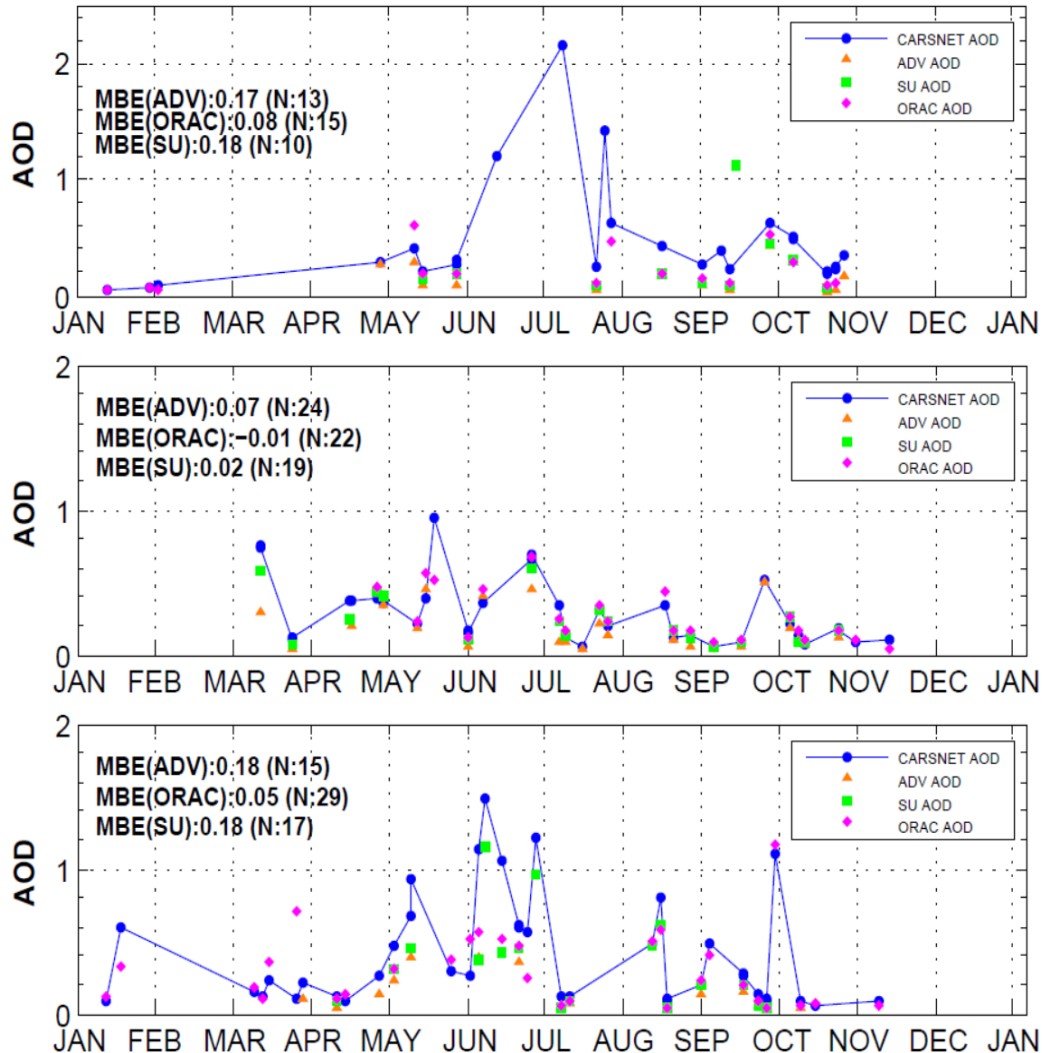

Fig. 6. Time series comparison of AATSR AOD with CARSNET AOD at Shangdianzi in 2007, 2008 and 2010.

Xianghe is located to the southeast of Beijing and has the same climatic conditions as Beijing. Approximately 98% of the surface is covered with urban land according to the MCD12C1 data of a 50 km × 50 km area. The performances of these three algorithms are at the same high quality level. However, the ADV algorithm still underestimated AOD at a level of MBE = 0.12 in 2007 and 0.10 in 2008.

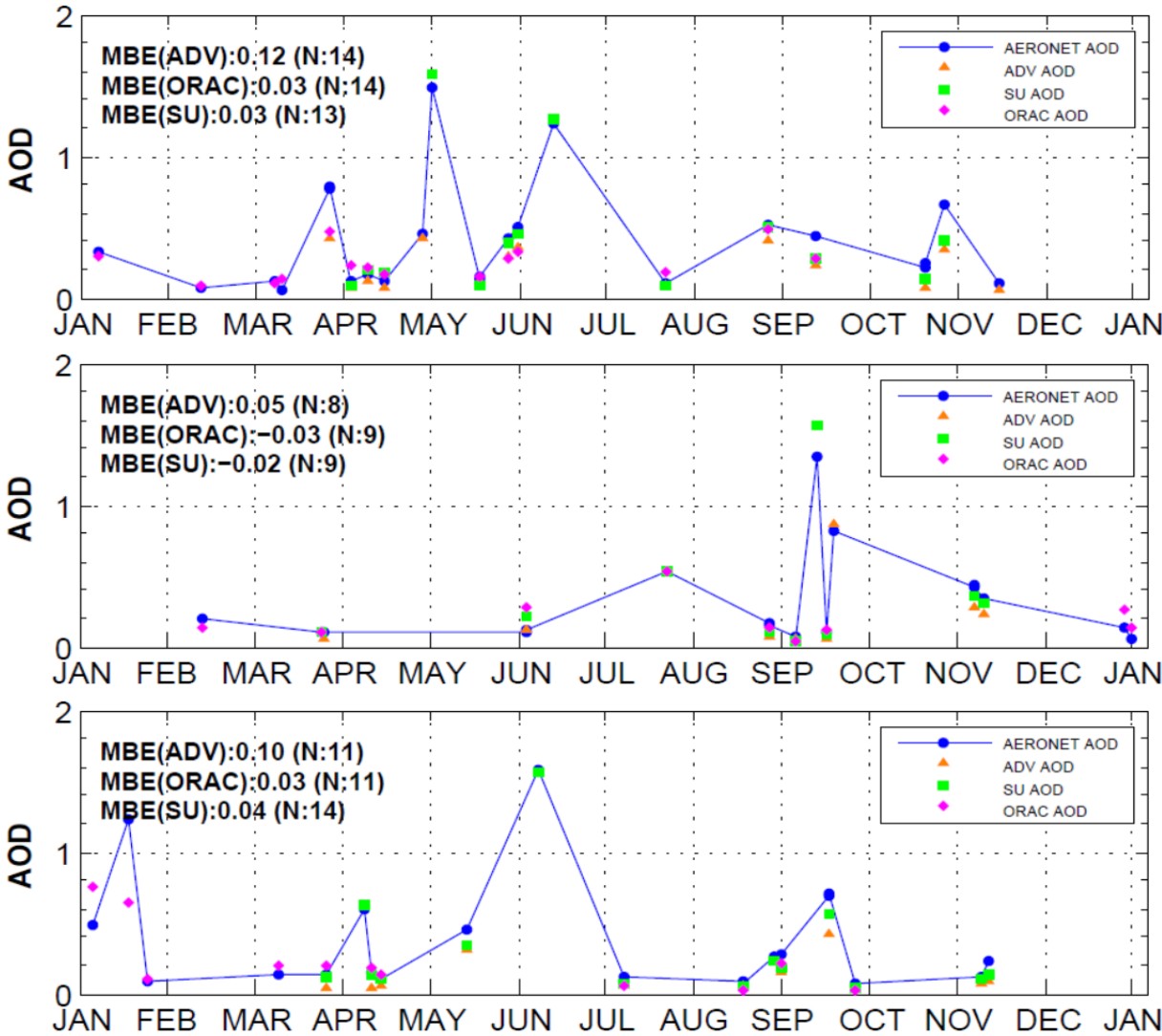

**Fig. 7. Time series comparison of AATSR AOD with AERONET AOD at XiangHe in 2007, 2008 and 2010.**

Xilinhot is situated at 43.95°N, 116.07°E, at the centre of the Xilinguole grassland. The main land cover is grassland (100%) based on the MODIS MCD12C1 data, with a spatial extent of 50 km × 50 km. The surface and climate features of Xilinhot are similar to those of SACOL, and the performance of the SU algorithm at these two sites is the same, i.e., both with low RMSE. The ADV algorithm slightly underestimated AOD with MBE of $0.10 \sim 0.13$. The ORAC AOD showed weak agreement

5    with the Xilinhot data, mainly because possible "outliers" exist in March to June 2008 and March 2010.

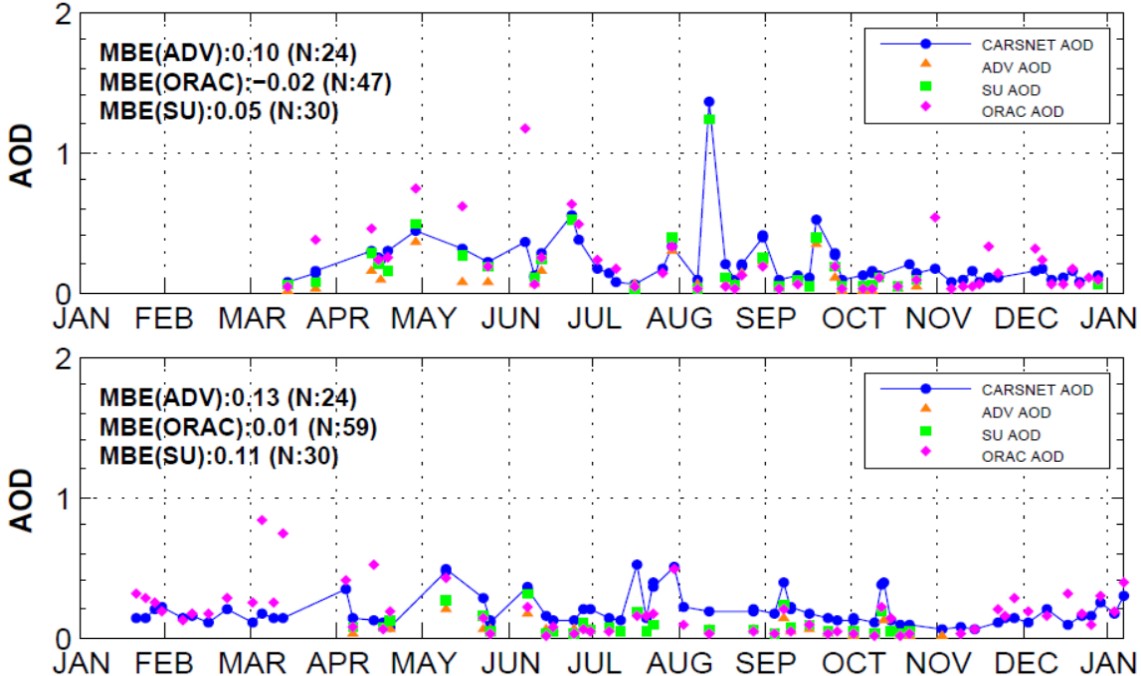

**Fig. 8. Comparison of SU AOD with CARSNET AOD at Xilinhot in 2008 and 2010.**

To guarantee statistical reliability, there must be more than 30 collocated pairs at one site. The determination of the surface cover at each site is based on the proportion (> 80% for one land type) of each land cover type from the MCD12C1 data at a

10    spatial extent of 50 km × 50 km. If no land cover type accounts for a proportion larger than 80% at a given site, it will be identified as mixed; then, we select two or more (sum > 80%) land types with the largest proportions as the main land cover. As the data volume is too low to infer the year-to-year variability of performance at these sites, the analysis gives some useful information but it is important not to over-interpret results from a small selection of data points.

**3.3.2 Analysis of algorithm performances in western China**

Because sufficient ground-based data in western China are lacking for the AERONET measurements, only data from CARSNET sites are used in 2008. We selected six CARSNET sites located in western China in which there are more than 25 matches.

5    Urumchi, situated at 43.78°N, 87.62°E, serves as the provincial capital of Xinjiang Uyghur Autonomous Region and is the most remote city in China in terms of distance to any sea. The dominant land cover at the spatial extent of 50 km × 50 km is grassland, which accounts for approximately 85%. The ADV, ORAC and SU algorithms all severely underestimated AOD, with MBE = 0.22, 0.12 and 0.17, respectively. The MBE is lowest mainly because of the "outlier" in April, which decreases the MBE.

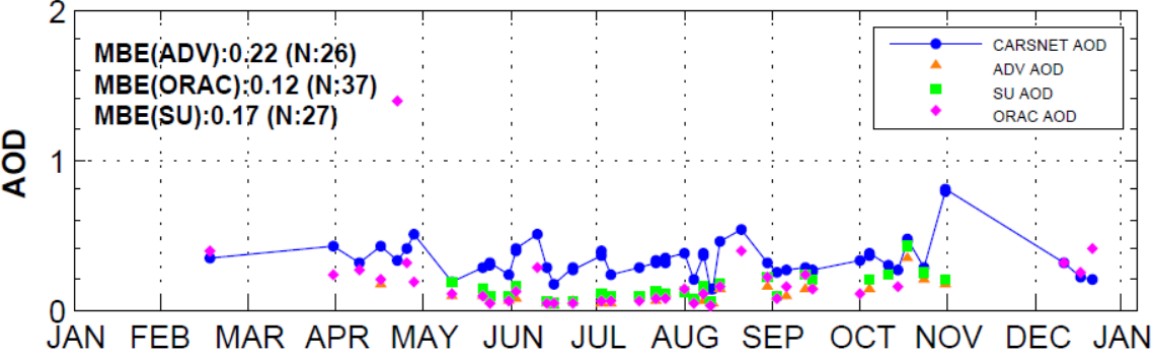

**Fig. 9. Time series comparison of AATSR AOD with CARSENT AOD at Urmchi in 2008.**

Ejina is situated at 41.95°N, 101.07°E, and its main land cover is barren ground (84%). The performances of ORAC and SU are at the same high quality level, with MBEs of 0.02 and 0.09, respectively. Another reason why we chose this site is that there are no matches of ADV products successfully collocated with ground-based data. Based on Fig. 10, the ORAC algorithm

15    has strong applicability in Ejina and high accuracy in retrieving AOD. The SU algorithm also performed well. This demonstrates that another limitation of the ADV algorithm is its applicability in calculating AOD in China. Dunhuang is situated at 40.15°N, 94.68°E and is surrounded by barren ground (85%). The same situation is true for Ejina, which causes slight underestimation at each point but high R and low RMSE for the ORAC algorithm (Figure 11). The performance of the SU algorithm was not as good as that of the ORAC because of its underestimation with MBE = 0.10. The limits of

20    underestimation and applicability of the ADV were more obvious at this site, as it only had 6 matches and showed severe underestimation with MBE = 0.17. Tazhong is situated at 39°N, 83.67°E and is surrounded by barren or sparsely vegetated

surface. Almost all land cover is barren ground according to the MODIS MCD12C1 data. Similar to the former two sites, the ADV product did not have any successful matches at this site (Figure 12). Both the ORAC and SU algorithms exhibited severe underestimation of retrievals, with MBE = 0.17 and 0.20, respectively. The outliers of the ORAC product in February are much higher than the observation data, causing the lower MBE.

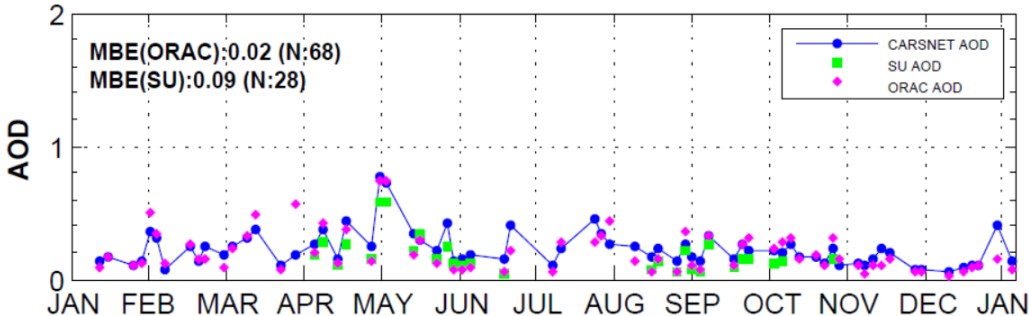

**Fig. 10. Time series comparison of AATSR AOD with CARSENT AOD at Ejina in 2008.**

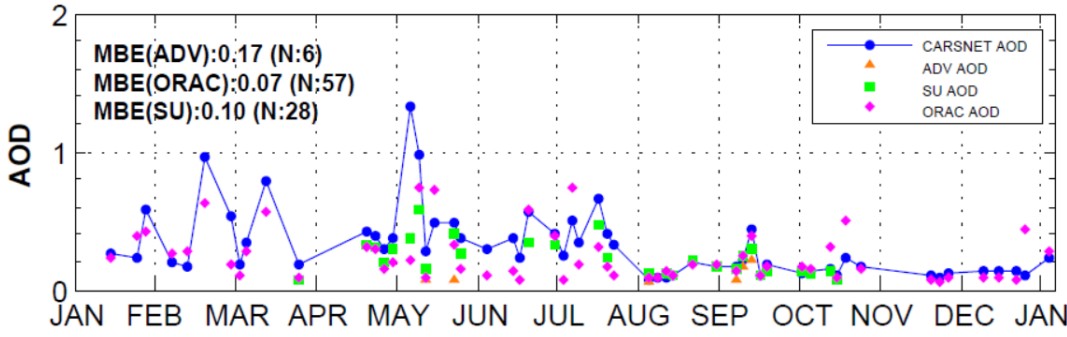

**Fig. 11. Time series comparison of AATSR AOD with CARSENT AOD at Dunhuang in 2008.**

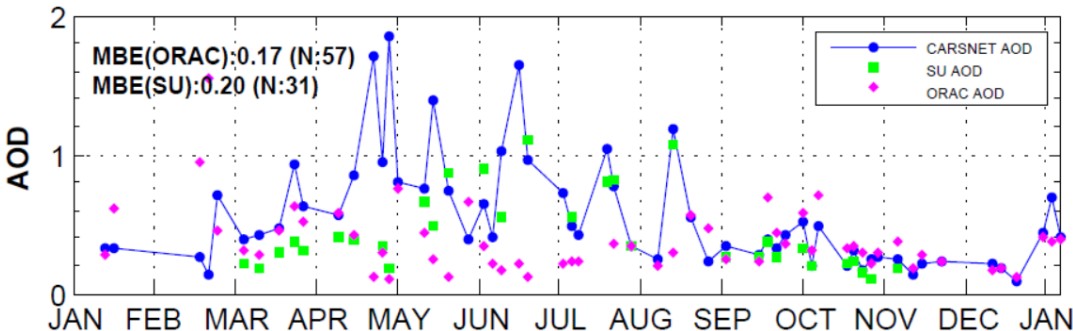

**Fig. 12. Time series comparison of AATSR AOD with CARSENT AOD over the site of Tazhong in 2008.**

The prevailing climatic pattern in western China is a temperate continental climate with four distinct seasons and less precipitation in winter and spring. In conclusion, compared to eastern China, the applicability of the ADV algorithm is not strong, and the underestimation is more severe. In the four selected sites in western China, the performance of the ORAC algorithm is best, even though severe underestimation occurs at some sites. The accuracy of the SU algorithm is not as high as
the ORAC product, with more severe underestimation and lower applicability.

### 3.3.3 Inter-comparison

In conclusion, the SU algorithm performs well in calculating AOD over different land covers from March to October. Slight underestimation occurs over barren ground or sparse vegetation at different times, and there are no obvious features in terms of precision in the time series over grasslands. For complex land surfaces where the dominant land cover is vegetation, the SU
algorithm is extremely effective in estimating AOD. In the last section, we draw a conclusion that the SU algorithm underestimates AOD over mainland China in 2008 probably because the dominant land cover in western China is barren or sparse vegetation, over which the SU algorithm underestimates AOD more severely.

The ADV algorithm underestimates AOD at most of the selected sites. We categorize these sites into four classes according to the MBEs of different sites: Class 1 (MBE<0.1), Class 2 (0.2>MBE>0.1), Class 3 (0.3>MBE>0.2), and Class 4 (MBE>0.3).
The ADV algorithm underestimates AOD over all selected sites, leading to all selected MBEs being larger than 0. We make such categories for the purpose of assessing the contribution of different surfaces to AOD estimation. Only XiangHe of 2008 belongs to Class 1, and Linan, Shangdianzi, and SACOL are classified into Class 2. Only Urumchi is in Class 3. Note that even though Lanzhou and Datong were not selected due their location, they should be classified in Class 4.

Overall, the ADV algorithm underestimates AODs at all sites but at different levels, as demonstrated by the above categories.
Serious underestimation occurs over the sites in Class 3 and Class 4 in western China, where the dominant land cover is a mixing of urban area and a large portion of grasslands. For the sites in Class 2, differences exist between Beijing and SACOL. SACOL is similar to the sites in Class 3 and Class 4, the main land cover of which is grassland. Over the sites in Class 1, the algorithm performs well with high R and low MBE, but there are no common features in terms of surface conditions.

The ORAC product collocates most pairs of all of these products. Most collocated pairs of the SU product and ADV product
occur in March to October, but the collocated pairs of the ORAC product occur during each month over some sites in 2008. Because more matches suggest greater errors for the determination of the "outlier" contribution to the overall performance of the ORAC algorithm, we introduce the ratio of the individual difference to average the differences for each site:

$$DR = \frac{\left|\tau_{AERO,i} - \tau_{sate,i}\right|}{\left(\sum_{i=1}^{n}\left|\tau_{AERO,i} - \tau_{Sate,i}\right|\right)}/n \qquad (6)$$

where DR<1 indicates a "relatively good" match, 3 >DR >1 indicates a "relatively poor" match, and DR >3 is an "outlier" (see Table 7).

There are no obvious "possible outliers" in Ejina shown in Fig. 10. Most of the DRs are in the range of 0 to 3, only two DRs are larger than 3, and the maximum (overestimation) is 5.112. The retrieved AOD in March is a possible "outlier" because it
is overestimated, whereas most are underestimated. Another two sites dominated by barren or sparsely vegetated land cover are Dunhang (approximately 85%) and Tazhong (100%). The conditions in Tazhong are complex, and there is no obvious relationship between the CARSNET data and the ORAC AODs. Most of the DRs are less than 3, and a total of eight DRs are larger than 3. The DR in February is an "outlier" because the varying tendencies are different between the ORAC product and the ground-based data, indicative of overestimation.

**Table 7. DR distribution of specific sites.**

| Site | DR<1 | 1<DR<3 | 3<DR<5 | 5<DR | Total |
|------|------|--------|--------|------|-------|
| Urumchi | 47 | 40 | 2 | 1 | 90 |
| Ejina | 51 | 43 | 1 | 1 | 96 |
| Tazhong | 63 | 17 | 5 | 3 | 88 |
| Dunhuang | 57 | 31 | 1 | 2 | 91 |

The ORAC product has the largest coverage at the expense of accuracy, especially in the presence of "outliers", and only the ORAC product has collocated validation pairs over some sites during each month in all three years. The ORAC algorithm underestimates AODs over Ejina, Tazhong and Dunhuang, but the "possible outliers" reduce the differences between the CARSNET data and the ORAC product. Xilinhot, Urumchi and SACOL share the same main land cover of grassland. The
problem is that the underestimations over these sites are not at the same level.

It is worth noting that the ORAC algorithm has the ability to calculate high AOD; however, most of the AODs have DRs larger than 3, indicating that the estimation of high AOD is unstable and has large error, reducing the overall precision.

**3.4 Seasonal characteristics of three algorithms**

The mainland China, cross about 60 degree of longitude and 30 degree of latitude, is dominated by monsoon-driven climate. In such vast territory, there are big differences in climate pattern from western to eastern China. The main climate type in eastern and eastern coastal China is monsoon climate. For western China far from the ocean, the climate type is hybrid of monsoon and continental climate. In dry seasons (winter, first half of spring, and last half of autumn), poor vegetation coverage,
loosen surface and winds in most northern China regions make coarse particles (sea salt and desert dust) into aerosol. Fine particles from coal combustion in winter and soot from straw burning in autumn is also important source of aerosol. In rainy seasons (mainly in summer), high vegetation blocks dust blowing into aerosol and reduce surface reflectance at visible wavelength. Table 8 shows the seasonal distribution of validation results of three algorithms. For the mainland China which is located in Northern Hemisphere from 20°N to 55°N, the spring time starts from about March to May, the summer time starts
from about June to August, the autumn time starts from about September to November and the winter time is about from December to February in next year.

**Table 8. Seasonal distribution of validation results of three algorithms.**

| | | N | MSA | MAA | MBE | MAE | RMSE | RMB | KAPPA |
|---|---|---|---|---|---|---|---|---|---|
| AATSR | Spring | 186 | 0.26 | 0.41 | -0.16 | 0.16 | 0.23 | 0.62 | 0.4 |
| ADV | Summer | 164 | 0.16 | 0.29 | -0.13 | 0.14 | 0.19 | 0.54 | 0.26 |
| | Autumn | 190 | 0.2 | 0.32 | -0.12 | 0.13 | 0.17 | 0.62 | 0.36 |
| AATSR | Spring | 294 | 0.35 | 0.37 | -0.02 | 0.18 | 0.3 | 0.95 | 0.5 |
| ORAC | Summer | 296 | 0.28 | 0.35 | -0.07 | 0.17 | 0.26 | 0.79 | 0.4 |
| | Autumn | 265 | 0.23 | 0.22 | 0.01 | 0.1 | 0.16 | 1.04 | 0.43 |
| | Winter | 230 | 0.29 | 0.28 | 0.01 | 0.15 | 0.25 | 1.03 | 0.38 |
| AATSR | Spring | 222 | 0.3 | 0.42 | -0.12 | 0.14 | 0.24 | 0.72 | 0.53 |
| SU | Summer | 241 | 0.32 | 0.43 | -0.11 | 0.13 | 0.2 | 0.74 | 0.49 |
| | Autumn | 237 | 0.26 | 0.35 | -0.09 | 0.11 | 0.16 | 0.74 | 0.49 |

Low mean of uncertainty (MUs) at 550nm means these retrievals are of high quality in Fig. 1. Most of Std_S are below 0.08, indicating high uniformity of ADV products (see Fig. 1). Most of collocated pairs of ADV AODs are concentrated below the
1-1 line and the RMB is 0.61, showing a tendency of underestimation. This kind of underestimation has an impact on ADV algorithm performances, for example, the RMS error is 0.19 in summer time, otherwise, the corresponding RMB is 0.54, which makes the KAPPA coefficient the smallest (0.26) than other seasons. The MBEs is from -0.12 in autumn to -0.16 in spring in Table 8, which means that the ADV algorithm tends to underestimate AOD in all seasons (except winter) over mainland China (See Figure 13). For monsoon climate, the main aerosol types in many parts of China are influenced by coarse particles (dust from Western China and sea salt from eastern coastal China) in spring time. The performance on calculating aerosol properties
of mixture of coarse particles is best in spring time with highest KAPPA coefficient, even though there are some samples with high MUs and the RMS error is 0.23.

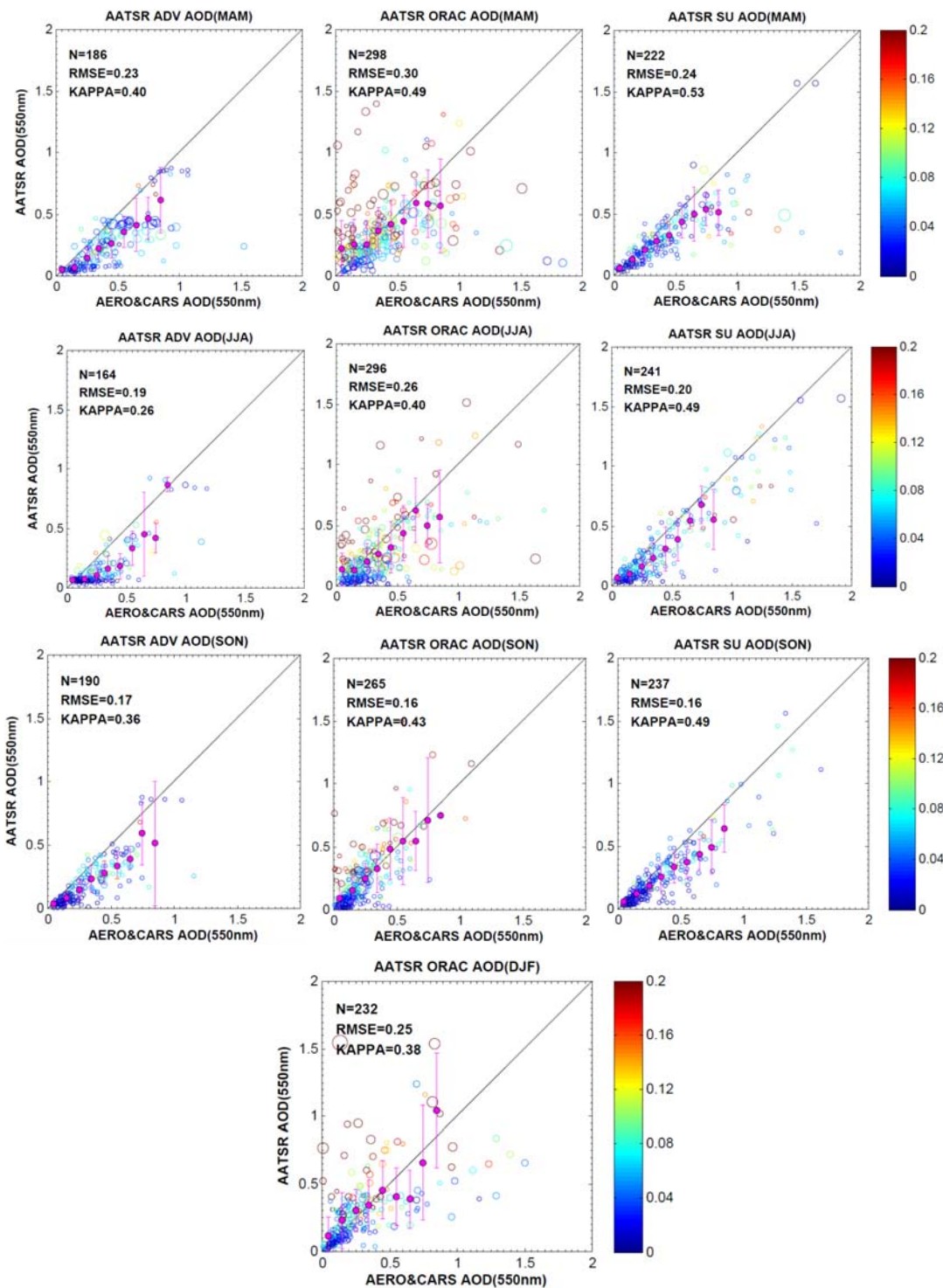

Fig. 13 Scatter plots of AATSR ADV, ORAC and SU L2 AOD products with ground-based data in China in the spring, summer, autumn and winter time of 2007, 2008 and 2010.

The matches of the ORAC product collocated with reference data are distributed discretely at two sides of 1-1 line in Fig. 1. The best performance with high KAPPA coefficient of 0.5 is in spring with no underestimation, even though the RMS error is high about 0.30. The KAPPA coefficient in the autumn time is lower than in the spring time, even though most of evaluation metrics are better in the autumn. Note that, only ORAC product of these three products has been collocated enough matches (more than 30) with reference data in the winter time. The performance of ORAC in winter is between that in spring and autumn without obvious underestimation or overestimation. The limitation of ORAC algorithm is the stability in retrieve aerosol properties, as shown in Fig. 13, the magenta mean $\pm 2\sigma$ lines for each season at each range are longer than those for other two products.

The SU algorithm has better performances in three years, getting KAPPA coefficients of 0.50. Most retrievals in matches are of high quality collocated with reference data and most Std_S are lower than 0.08, i.e. the sample quality is high and this coincides with assumption of aerosol properties uniformity in 50km × 50km area. The best performance on retrieval is in the autumn, lowest RMS error of 0.16 and largest RMB of 0.74 in three seasons shown in Fig. 13. The magenta lines are similar with those of ADV product in corresponding seasons, showing same level of stability in retrieving AOD. The SU algorithm has no obvious differences in retrieving AOD in three seasons. One limitation of SU and ADV algorithms is less than 30 collocated matches in the winter time so that we can't evaluate its performance during that time.

The latest MODIS Moderate Resolution Imaging Spectroradimeter (MODIS) Collection 6 (C6) product were released in 2013, including aerosol datasets produced by two "Dark Target" (DT) algorithms (one is for retrieving over ocean and the other is for retrieving over land) and "Deep Blue" algorithm for retrieving over bright or semi-arid surface (Levy et al., 2013). For over land, the DT algorithm uses an updated cloud mask to allow retrieval of heavy aerosol compared to algorithm employed in MODIS Collection 5. It is reported that MODIS C6 products (produced by three algorithms) are of high quality (Sayer at el., 2014). Here, we select both MODIC C6 DT and DB 10km × 10km merged dataset as reference data for cross-validation of AATSR L2 AOD products. The matches in Fig. 14 are randomly chosen from MODIS and AATSR collocated AOD datasets. The ADV AOD has lowest RMSE of 0.11. The SU algorithm has same performance with ORAC (similar RMSE and KAPPA) but with a little underestimation as the magenta line in Fig. 14.

Aerosol Angstrom Exponent is an exponent that expresses the spectral dependence of aerosol optical thickness with the wavelength of incident light (Eck et al. 1999). The Ångström exponent is inversely related to the average size of the particles in the aerosol: the smaller the particles, the larger the exponent. Thus, Ångström exponent is a useful quantity to assess the particle size of atmospheric aerosols or clouds, and the wavelength dependence of the aerosol/cloud optical properties.

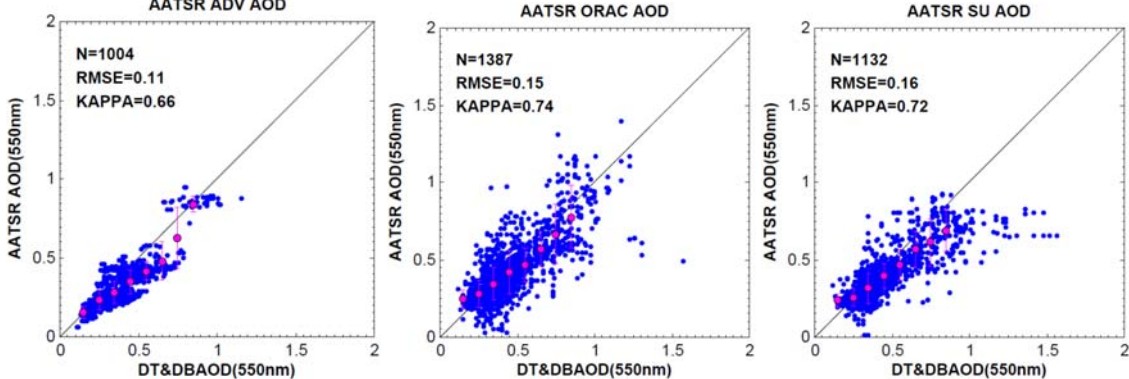

**Fig. 14 Scatter plot of AATSR AOD and DT&DB AOD**

The ORAC product provides Ångström exponent for 550-870μm only and SU product provides Ångström exponent for 550-870μm only. The CARSNET dataset provides Ångström exponent for 440-870μm only. As the ADV product provides Ångström exponent for 550-670μm only, we couldn't do comparison for ADV Ångström exponent. We compared Ångström exponent using both CARSNET and AERONET datasets for SU and ORAC products. Figure 15 shows the comparisons of Ångström exponent. In general, both SU and ORAC algorithms generate similar quality of Ångström exponent values. There is no any pattern of Ångström exponent with AOD values and uncertainty.

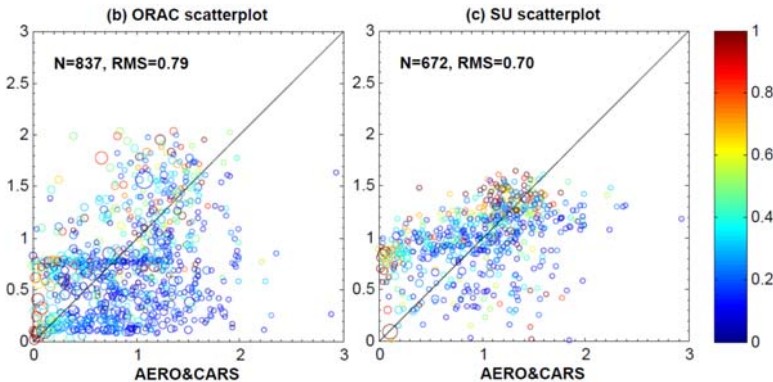

**Fig. 15 Comparisons of Ångström exponent of ORAC and SU products. The area and colours of bubbles represent AOD uncertainty and AOD values, respectively.**

## 4. Conclusions

Satellite remote sensing of atmosphere is one important aspect in the PEEX scientific plan. Remotely-sensed data could provide continuous spatial coverage of aerosol property over the Pan-Eurasian area. These three algorithms (the SU algorithm, the ADV algorithm and the ORAC algorithm) display different performances in estimating AOD over mainland China in 2007, 2008 and 2010. However, none of the algorithms show an explicitly better performance than the other two. The SU and ADV products have higher accuracy over most selected sites but less coverage, whereas the ORAC product has greater coverage at the cost of accuracy.

All of these algorithms tend to underestimate AOD to some degree. The underestimation becomes more severe with increasing AOD or aerosol loading. The method of grouping helps to identify "possible outliers" in different regions of aerosol loading.

The precision of the SU and ADV algorithms is at the same level over different surfaces. However, the SU product has more strict quality control than the ADV product, and it eliminates AODs to make the MBE less than 0.10 over different sites (de Leeuw et al. 2013). Over grassland and barren vegetation, the SU displays a strong performance with slight underestimation (MBE < 0.10). The limitations of underestimation and applicability of the ADV are more obvious over such sites. For complex surfaces with two or more land cover types, the performances of these three algorithms are at the same level. Note that Lanzhou and Datong are different from other sites, even though the main land cover type is grassland. All of these algorithms underestimated AOD at a high level, perhaps because these algorithms are not sensitive to absorptive aerosols.

Only the ORAC product shows "possible outliers" identified by equation (2), which substantially decreases its accuracy. The most obvious feature of the "possible outliers" is that the retrieved AODs are higher than the ground-based measurements.

As reference data, AERONET L2 data have some limitations, including the distribution and number of sites in mainland China. Most sites of AERONET are distributed in eastern China and the coastal region of China for special experimental use; as a result, sufficient reference data cannot be obtained to validate the AOD product. The CARSNET data make up for this shortage because there are more CARSNET sites in China, especially in western China, where few AERONET sites have been constructed. Limited both by reference data and satellite retrievals, most co-allocated pairs occur in March to November, and few occur in winter (December, January and February).

## Acknowledgements

This work was supported in part by the Ministry of Science and Technology (MOST) of China under Grant Nos. 2013CB733403 and 2013AA122801, the National Natural Science Foundation of China (NSFC) under Grant Nos. 41471306

and GF 30-Y20A02-9003-15/16, and the EU/FP7 MarcPolo project (Grant Agreement No. 606953). Part of the work was conducted in preparation for the Aerosol_cci project (ESA-ESRIN project AO/1-6207/09/I-LG), from which three AATSR AOD products were provided. The data for uncertainty analysis and validation came from 34 AERONET sites and eight CARSNET sites. We thank the PIs, investigators and their staff for establishing and maintaining the data for this study.

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
