# Peer review of "Inter-comparison of three AATSR Level 2 (L2) AOD products over China"

_Atmospheric Chemistry and Physics, 2016_

## Referee Comment (RC1) · Anonymous Referee #1 · 25 Mar 2016

In this study, the authors validated three AATSR AOD products (ADV, ORAC and SU algorithm) provided by Aerosol_cci project over China in 2007, 2008 and 2010. It has been widely validated (compared with AERONET AOD) that these three algorithms have ability in retrieving AOD over land globally with high precision. However, over China mainland area, the AERONET data has limitations as reference data. There were not enough AERONET sites built and the distribution of AERONET sites were unevenly in mainland China because of large territory of mainland China. The authors introduced CARSNET data to be combined with AERONET data, making up for these limitations and improving reliability of reference data. On this basis, the authors not only select common evaluation metrics, but also introduce new metrics, for example, the improved KAPPA coefficient as comprehensive evaluation metric, the DR for determination of AOD retrieved "outliers", the improved expected error envelope designed for characteristics of AATSR AOD products, etc. This study is a nice trial consisting many meaningful works and I would recommend publication if my following comments/suggestions can be adequately addressed.

Major comments:
1. The structure and composition of manuscript should follow the requests of official website of Atmospheric Chemistry and Physics (ACP). For example, keywords, team list, etc. should add to manuscript and team list exist in this manuscript.

2. Figures in the manuscript should be clear and easily understood. The main method of this study is to validate three AATSR AOD products year by year for reason of different reference data available for the authors. Readers could distinguish which sites in the Fig. 1 are from AERONET or CARSNET, but may not pick out the space distribution of ground-based data sites in same year easily. I recommend authors to replot Fig. 1 of "The distribution of selected AERONET&CARSNET sites in mainland China in 2007, 2008 and 2010", using one same color or type for sites available in one year.

3. Also I suggest that the paper never use the word "good" to describe the results. The coefficient of correlation (CC) as one of main evaluation metric, which indicates whether there is any linear relationship among the points. Authors could not claim which performances of products is "good" or not "good" by any values of CC or other evaluation metrics. For example, when CC is high, the performances could be viewed as "good", when CC is low, the performance is also viewed "good". The word "good" may confuse readers, leading misunderstanding of conclusions in this study.

Specific comments:

Page 2 line 9, the influences of aerosol particles on cloud should cited the paper of Twomey published in 1974. In general some more references should be added in lines 5-12.

Page 2 line 19, the word "because" should be replaced by other words like "including"

Page 3 line 10, the word "more" should be removed

Page 3 line 15-19, comparison of satellite retrievals with other high quality has limitations, could you illustrate it more clearly?

Page 3 line 26, Please use "Aerosol_CCI" or "Aerosol_cci" through the manuscript but not the mix of them?

Page 4 Tab. 1 the bottom row are same with header row, what is it useful? And in the title abbreviation "Tab." should avoid.

Page 9 Tab. 4 these statistics should be up to two decimal point.

Page 9 line 3, this sentence has syntax error.

Fig. 8 – Fig. 16. The places of titles should be same.

Page 19 Tab. 5 these statistics should be up to two decimal point.

Page 24 line 29, in part of acknowledgements, the numbers of sites are inconsistent with mentioned as above.

---

## Referee Comment (RC2) · Anonymous Referee #2 · 25 Mar 2016

This paper validates 3 algorithms (SU, ADV, ORAC) for determining AOD from the European AATSR sensor against Sun photometer data in China (from AERONET and CARSNET). The topic is relevant to ACP. The work is important because these European products have not been as well known as NASA ones, and have undergone a lot of development in the European CCI projects, so it is a good time to do some more thorough validation of these data sets. This is especially true for China since the aerosol loading is high and variable, and CARSNET has monitoring stations in some areas where AERONET is lacking.

I have read through the manuscript several times and, while it is promising, there are some things which are unclear/invalid or I think not useful, and some important things which should be added to make the analysis more complete and useful. The phrasing is odd in some places and there are a number of typos (e.g. AEROENT in some places

instead of AERONET) so I think that the manuscript will need some copy editing by the production office. I appreciate that English is not the authors' first language and the writing is not bad, it is just unclear in some cases. I therefore recommend some content revisions, to address the points below. I would like to review a revised version and think that another round of reviews will be necessary because the structure/content of the manuscript might change a lot. Here are my main points:

1. Abstract: some of the sentences could probably be shortened (e.g. first and second can be combined, as can third and fourth).

2. Introduction and start of section 2: It would be good to add a bit more information about the AATSR sensor here, like launch/end dates, swath width. A brief discussion of the differences between the algorithms should be included, to help understand why they give different results. From the analysis, the performance and the spatial coverage are both different between the three algorithms, so some insight into what in the algorithms is responsible would be welcome.

3. Statistics. Some of the metrics presented here are questionable in their relevance and I think that there are simpler and clearer alternatives. Specifically, the EE envelopes quoted here for Equations 1, 2 are for the MODIS instrument, not AATSR. AATSR is quite different (two views, fewer wavelengths) so there is no reason to expect that an AOD retrieval for AATSR would have the same type of behaviour. One might expect that the error formulation would be closer to that of MISR. Further, Equations 3 and 4 are basically expressing a confidence envelope around a regression line. This is not really useful since it is just the noise around the relationship and not so dependent on the actual error in the retrievals. So comparing this between algorithms does not really make sense. A well-correlated but very biased retrieval would appear 'better' by this metric than a poorer-correlated but unbiased one, while for an actual scientific application, the unbiased one may in some cases be more useful.

Further, least-squares linear regression is invalid for AOD retrievals because aerosol data violate the assumptions of this technique (see e.g. http://people.duke.edu/~rnau/testing.htm ; the AOD data validate assumptions 3 and 4 that linear least-squares regression makes, possibly 1 and 2 as well, and as a result the results obtained are not statistically valid). I know that a lot of people do least-squares linear regression because it is easy, but it is still wrong for this application.

So, a better alternative is just to present statistics of bias and RMS error as a function of AOD, similar to what is shown in e.g. Figure 5 and the magenta bins in Figure 2. So I suggest that the EE discussion here and linear regressions be discarded entirely, and more prominence should be given to statistics subset into different regimes (e.g. low AOD, moderate AOD, high AOD; perhaps also splits based on Angstrom exponent for the high-AOD regime, as retrieval errors are often type-dependent as well). The kappa coefficient is probably fine. So, accounting for this comment would somewhat streamline and improve sections 2 and 3.

Finally, presenting statistics to 3 significant figures is overkill and paints a picture of them being more robust than they probably are; 2 significant figures is probably enough.

4. Retrieval errors. As I understand it, the CCI project means that the data products also provide their own estimates of the uncertainty in the AOD retrieval for every pixel. This is an important point, since pixel-level uncertainties are very useful for many applications. However, it is not discussed in the paper. How do these uncertainty estimates compare to the retrieval errors observed?

5. Figures 2-4 and discussion in Section 3. I don't see any advantage to splitting out the points by year. It would be easier to combine all points from one algorithm into one panel, not 3. This would also let you combine Figures 2-4 into one figure for a side-by-side comparison of the three algorithms. Also, as discussed before, I would delete the regression and EE lines here since they are not very meaningful. The magenta symbols and lines for the binned data are enough here. Also, the color scales used in

these figures are not mentioned and can probably be removed (either show individual points without a color scale or a density plot with a color scale).

I also don't see any good reason to split the discussion of statistics out by year either. The data volume is not very large, so year to year differences are probably resulting from sampling and not statistically meaningful. Looking at the bigger picture of all data together is more statistically robust and gives a clearer picture. I don't believe any insight is gained by splitting the analysis up year by year.

6. Figures 5-7: Similar to the last comment: are the different panels the different years? It doesn't say anywhere but I infer that is the case. Again, these figures could be streamlined into one because clearly the biases are similar between years, this will be more robust, and will allow for a more direct comparison of the 3 algorithms. Additionally, I don't think the histograms (bottom panels) here are useful since they don't provide any information which is not seen clearly in the top panels, so these could be deleted. Also, for the same reason as before, the linear regressions are invalid and should be deleted, just showing the binned values is enough.

7. A similar plot to the bias plots could be created for RMSE. This would be a clearer way to show and compare the AOD-dependence of the retrieval error than the EE3/EE4 metrics.

8. In the discussion of the results, a lot of the time terms like "good" and "well" are used to describe performance. These are "weasel words" and should be avoided. What is "good" is only really relevant relative to a specific application (e.g. good enough to do X) or compared to the state of the art. I suggest rewording to avoid these words and be more quantitative where possible, or else stick to comparative terms (e.g. say when the data sets are similar to or better than each other). Also, some discussion of results compared to validation of other sensors (the main ones being MODIS/MISR) could be included, as these all have published validation for their aerosol products, and this would give a sense of how the AATSR data perform relative to the other available

data products. Right now the paper more or less reads like AATSR is the only satellite option.

9. Do the retrievals provide other information like Angstrom exponent? From other references, I believe so. This quantity is commonly compared with AERONET measurements, so it should be easy to extend the analysis to look at this as well using the same basic approaches. This might provide more insight for the differences between the data sets, if the algorithms are making very different assumptions about what sort of aerosol is present. This would help overcome one of the weaknesses of the paper, i.e. that the comparison is presented without any sort of discussion about why the three data sets are different and how to improve them (which would be very useful information).

10. There are at least two more AERONET sites in China which provided data in the study period, but which were not used in the analysis. These are both in Hong Kong: Hong_Kong_Hok_Tsui and Hong_Kong_PolyU. Why were these not used? If the objective (as stated) is to provide coverage over broad areas of China, then it would make sense to include them, since the data are freely available and there are no other sites used in this part of China. These sites are very close to the coast so also provide an additional type of environment to analyse, compared to the other sites presently included in the study. Additionally, it will boost the data volume. I suggest adding these sites to the analysis. There may be more, these were the main ones which sprang to mind. On a related note, Figure 1 can probably be simplified for clarity by using one symbol/color for all AERONET sites, and another for all CARSNET sites. Splitting by year isn't necessary, in my view, and just complicates things.

11. The title of the manuscript suggests a broader scope than the analysis, since the analysis only performs an inter-comparison in the context of AERONET/CARSNET measurements. There are various other things which could be added, at least briefly. For example, climatologies of seasonal AOD from all three algorithms (from the 1 degree products), and maps showing the available data volume (e.g. number of days per

season with data), since this is another feature which is important for many applications. Otherwise, the title should be amended to reflect the scope. However I would prefer that the analysis be extended because I think that this would be quite useful (and new, to my knowledge).

---

## Short Comment (SC1) · 4 May 2016

Dear Editor and Reviewers,

We highly appreciate the detailed valuable comments from the referees on our manuscript of "acp-2016-195". The suggestions are quite helpful and we have incorporated them in the new version of manuscript. We have referred to literatures and papers and re-analyzed the collected data and reconstructed the paper to improve the quality of our paper.

As below, I would like to clarify some of the points raised by the Reviewers. And we hope that the reviewers and the editors will be satisfied with our responses to the 'comments' and the revisions for the original manuscript.

[Figure]

Yours truly,

Yong Xue

  In this study, authors validate three AATSR AOD products (ADV, ORAC and SU algorithm) provided by Aerosol_cci project over China in 2007, 2008 and 2010. It's been widely validated (compared with AERONET AOD) that these three algorithms have ability in retrieving AOD over land with high precision till 3/23/2016. However, the AERONET data has limitations as reference data that there were not enough AERONET sites built and the distribution of AERONET sites were unevenly in mainland China in 2007, 2008 and 2010 caused by large territory of mainland China. Authors introduce CARSNET data to be combined with AERONET data, making up for these limitations and improving reliability of reference data. On this basis, authors not only select common evaluation metrics, but also introduce new metrics, for example, the improved KAPPA coefficient as comprehensive evaluation metric, the DR for determination of AOD retrieved "outliers", the improved expected error envelope designed for characteristics of AATSR AOD products, etc. This study is a nice trial consisting many meaningful works and I would recommend publication if my following comments/suggestions can be adequately addressed.

Major comments: 1. The structure and composition of manuscript should follow the requests of official website of Atmospheric Chemistry and Physics (ACP). For example, keywords, team list, etc. should add to manuscript and team list exist in this manuscript. Response: All required structure and composition will be added in new version of manuscript.

2. Figures in the manuscript should be clear and easily understood. The main method of this study is to validate three AATSR AOD products year by year for reason of different reference data available for authors. Readers could distinguish which sites in the Fig. 1 is from AERONET or CARSNET, but may not pick out the space distribution of ground-based data sites in same year easily. I recommend authors replot Fig. 1 of "The

distribution of selected AERONET&CARSNET sites in mainland China in 2007, 2008 and 2010", using one same color or type for sites available in one year. Response: The figures have been reploted with clear and easy understood symbols and text, as Fig. 1 we have reploted the symbols of sites from different networks using different colors to make it clearer.

3. Also I suggest that the paper never use the word "good" to describe the results. The coefficient of correlation (CC) as one of main evaluation metric, which indicates whether there is any linear relationship among the points. Authors could not claim which performances of products is "good" or not "good" by any values of CC or other evaluation metrics. For example, when CC is high, the performances could be viewed as "good", when CC is low, the performance is also viewed "good". The word "good" may confuse readers, leading misunderstanding of conclusions in this study. Response: The "good" or "well" terms have been replaced by quantitative description or comparative words. For example, in section 4 to section 6, we have refined our analysis using more detailed quantitative description to present readers easy understood analysis.

Specific comments:

Page 2 line 9, the influences of aerosol particles on cloud should cited the paper of Twomey published in 1974. Response: This reference will be added.

Page 2 line 19, the word "because" should be replaced by other words like "including" Response: This sentence has been revised.

Page 3 line 10, the word "more" should be removed Response: This word has been removed.

Page 3 line 15-19, comparison of satellite retrievals with other high quality has limitations, could you illustrate it more clearly? Response: We have added necessary illustration and cross-validation with MODIS C6 DT&SB merged datasets.

Page 3 lin26, is "Aerosol_CCI" or "Aerosol_cci" formal? Response: The CCI official website uses "Aerosol_cci", therefore, we'll introduce "Aerosol_cci" in the following or revised paper.

Page 4 Tab. 1 the bottom row are same with header row, what's it useful? And in the title abbreviation "Tab." should avoid. Response: Tab. 1 has been revised.

Page 9 Tab. 4 these statistics should be up to two decimal point. Response: Relative statistics have been kept two places of decimal.

Page 9 line 3, this sentence has syntax error. Response: This sentence has been revised in new version of manuscript.

Fig. 8 – Fig. 16, the places of titles should be same. Response: The places of title in figures have been adjusted to the same.

Page 19 Tab. 5 these statistics should be up to two decimal points. Response: Relative statistics have been kept two places of decimal as Tab. 4.

Page 24 line 29, in part of acknowledgements, the numbers of sites are inconsistent with mentioned as above. Response: The numbers of ground-based sites have been corrected in new version of manuscript.

Please also note the supplement to this comment:
http://www.atmos-chem-phys-discuss.net/acp-2016-195/acp-2016-195-SC1-supplement.pdf
* * *

---

## Short Comment (SC2) · Anonymous Referee #2 · 4 May 2016

Dear Editor and Reviewers,

We highly appreciate the detailed valuable comments from the referees on our manuscript of "acp-2016-195". The suggestions are quite helpful and we have incorporated them in the revised paper. We have referred to literatures and papers and re-analyzed the collected data and reconstructed the paper to improve the quality of our paper.

As below, I would like to clarify some of the points raised by the Reviewers. And we hope that the reviewers and the editors will be satisfied with our responses to the 'comments' and the revisions for the original manuscript.

[Figure]

Yours truly,

Yong Xue   Interactive comment on "Inter-comparison of three AATSR Level 2 (L2) AOD products over China" by Y. Che et al.

Anonymous Referee #2

This paper validates 3 algorithms (SU, ADV, ORAC) for determining AOD from the European AATSR sensor against Sun photometer data in China (from AERONET and CARSNET). The topic is relevant to ACP. The work is important because these European products have not been as well-known as NASA ones, and have undergone a lot of development in the European CCI projects, so it is a good time to do some more thorough validation of these data sets. This is especially true for China since the aerosol loading is high and variable, and CARSNET has monitoring stations in some areas where AERONET is lacking.

have read through the manuscript several times and, while it is promising, there are some things which are unclear/invalid or I think not useful, and some important things which should be added to make the analysis more complete and useful. The phrasing is odd in some places and there are a number of typos (e.g. AEROENT in some places instead of AERONET) so I think that the manuscript will need some copy editing by the production office. I appreciate that English is not the authors' first language and the writing is not bad, it is just unclear in some cases. I therefore recommend some content revisions, to address the points below. I would like to review a revised version and think that another round of reviews will be necessary because the structure/content of the manuscript might change a lot. Here are my main points:

Response: The English of the manuscript has been edited by the Elsevier's Language Services.

1. Abstract: some of the sentences could probably be shortened (e.g. first and second

can be combined, as can third and fourth).

Response: The sentences have been shortened and refined in new version of abstract.

2. Introduction and start of section 2: It would be good to add a bit more information about the AATSR sensor here, like launch/end dates, swath width. A brief discussion of the differences between the algorithms should be included, to help understand why they give different results. From the analysis, the performance and the spatial coverage are both different between the three algorithms, so some insight into what in the algorithms is responsible would be welcome.

Response: More details and information about the AATSR instrument and retrieval algorithms have been added, furthermore, a brief analysis and discussion of the differences between the retrieval algorithms have also been added in the revised version of manuscript. These information will help readers to have a deep insight about the differences of validation result we have made in this study.

3. Statistics. Some of the metrics presented here are questionable in their relevance and I think that there are simpler and clearer alternatives. Specifically, the EE envelopes quoted here for Equations 1, 2 are for the MODIS instrument, not AATSR. AATSR is quite different (two views, fewer wavelengths) so there is no reason to expect that an AOD retrieval for AATSR would have the same type of behavior. One might expect that the error formulation would be closer to that of MISR. Further, Equations 3 and 4 are basically expressing a confidence envelope around a regression line. This is not really useful since it is just the noise around the relationship and not so dependent on the actual error in the retrievals. So comparing this between algorithms does not really make sense. A well-correlated but very biased retrieval would appear 'better' by this metric than a poorer-correlated but unbiased one, while for an actual scientific application, the unbiased one may in some cases be more useful. Further, least-squares linear regression is invalid for AOD retrievals because aerosol data violate the assumptions of this technique (see e.g. http://people.duke.edu/âĹijrnau/testing.htm ; the AOD

data validate assumptions 3 and 4 that linear least-squares regression makes, possibly 1 and 2 as well, and as a result the results obtained are not statistically valid). I know that a lot of people do least-squares linear regression because it is easy, but it is still wrong for this application. So, a better alternative is just to present statistics of bias and RMS error as a function of AOD, similar to what is shown in e.g. Figure 5 and the magenta bins in Figure 2. So I suggest that the EE discussion here and linear regressions be discarded entirely, and more prominence should be given to statistics subset into different regimes (e.g. low AOD, moderate AOD, high AOD; perhaps also splits based on Angstrom exponent for the high-AOD regimeïijĽ, as retrieval errors are often type-dependent as well). The kappa coefficient is probably fine. So, accounting for this comment would somewhat streamline and improve sections 2 and 3. Finally, presenting statistics to 3 significant figures is overkill and paints a picture of them being more robust than they probably are; 2 significant figures is probably enough.

Response: One objective of this manuscript is to evaluate different statistical metric for the validation of quantitative remote sensing. Different statistical metric shows different meaning and is used for different purpose. Linear regression is the most basic and commonly used statistical method that allows us to summarize and study relationships between two quantitative variables. Pearson correlation coefficient (CC) measures the fraction of the total variability in the response that is accounted for by the retrieval and is only a measure of linear association between ground truth measurements and satellite retrieval values. Bias describes the average difference between satellite retrievals and ground AOD. For the consistency of the metric among different aerosol products, it is better to show the percent of retrievals falling within the expected error (EE) range.

4. Retrieval errors. As I understand it, the CCI project means that the data products also provide their own estimates of the uncertainty in the AOD retrieval for every pixel. This is an important point, since pixel-level uncertainties are very useful for many applications. However, it is not discussed in the paper. How do these uncertainty estimates compare to the retrieval errors observed?

Response: The satellite retrieved AOD in each collocated pair are means of retrievals in 5 × 5 sampling frame. On this basis, we calculate means of uncertainty estimates in sampling area for each collocated pair as sizes of circles in scatter plot. In section 4 and 5, we reanalyze validation results of different algorithms, including comparison of uncertainty estimates and retrievals error observed.

5. Figures 2-4 and discussion in Section 3. I don't see any advantage to splitting out the points by year. It would be easier to combine all points from one algorithm into one panel, not 3. This would also let you combine Figures 2-4 into one figure for a side-by-side comparison of the three algorithms. Also, as discussed before, I would delete the regression and EE lines here since they are not very meaningful. The magenta symbols and lines for the binned data are enough here. Also, the color scales used in these figures are not mentioned and can probably be removed (either show individual points without a color scale or a density plot with a color scale). I also don't see any good reason to split the discussion of statistics out by year either. The data volume is not very large, so year to year differences are probably resulting from sampling and not statistically meaningful. Looking at the bigger picture of all data together is more statistically robust and gives a clearer picture. I don't believe any insight is gained by splitting the analysis up year by year.

Response: All points from one algorithm have been combined to make results more statistically robust and remove unnecessary plots. The colors of points in new scatter plot represent standard deviation of retrievals in sampling area for the purpose of finding influence on retrieving performance of sampling. We also keep the comparisons for each year as we would like to see the differences for each year. We added one section on the analysis of seasonal behaves of three algorithms.

6. Figures 5-7: Similar to the last comment: are the different panels the different years? It doesn't say anywhere but I infer that is the case. Again, these figures could be streamlined into one because clearly the biases are similar between years, this will be more robust, and will allow for a more direct comparison of the 3 algorithms.

Additionally, I don't think the histograms (bottom panels) here are useful since they don't provide any information which is not seen clearly in the top panels, so these could be deleted. Also, for the same reason as before, the linear regressions are invalid and should be deleted, just showing the binned values is enough.

Response: New scatter plots have been made, combining all points from one algorithm.

7. A similar plot to the bias plots could be created for RMSE. This would be a clearer way to show and compare the AOD-dependence of the retrieval error than the EE3/EE4 metrics.

Response: RMS error has been added in plot and statistic table.

8. In the discussion of the results, a lot of the time terms like "good" and "well" are used to describe performance. These are "weasel words" and should be avoided. What is "good" is only really relevant relative to a specific application (e.g. good enough to do X) or compared to the state of the art. I suggest rewording to avoid these words and be more quantitative where possible, or else stick to comparative terms (e.g. say when the data sets are similar to or better than each other). Also, some discussion of results compared to validation of other sensors (the main ones being MODIS/MISR) could be included, as these all have published validation for their aerosol products, and this would give a sense of how the AATSR data perform relative to the other available data products. Right now the paper more or less reads like AATSR is the only satellite option.

Response: The "weasel words" like "good" or "well" have been replaced by details of RMSE and KAPPA coefficient or comparative words. We compare and analyze AATSR AOD with "Deep Blue" and "Dark Target" 10km×10km AOD data from MODIS Collection 6 datasets which has been widely validated.

9. Do the retrievals provide other information like Angstrom exponent? From other

references, I believe so. This quantity is commonly compared with AERONET measurements, so it should be easy to extend the analysis to look at this as well using the same basic approaches. This might provide more insight for the differences between the data sets, if the algorithms are making very different assumptions about what sort of aerosol is present. This would help overcome one of the weaknesses of the paper, i.e. that the comparison is presented without any sort of discussion about why the three data sets are different and how to improve them (which would be very useful information).

Response: The CARSNET dataset provides AOD and angstrom exponent (440-870) only, otherwise the ADV provides angstrom exponent (550-670) only, ORAC provides angstrom exponent (550-870) only and SU provides angstrom exponent (550-870) only. Comparison between these data may be invalid.

10. There are at least two more AERONET sites in China which provided data in the study period, but which were not used in the analysis. These are both in Hong Kong: Hong_Kong_Hok_Tsui and Hong_Kong_PolyU. Why were these not used? If the objective (as stated) is to provide coverage over broad areas of China, then it would make sense to include them, since the data are freely available and there are no other sites used in this part of China. These sites are very close to the coast so also provide an additional type of environment to analyze, compared to the other sites presently included in the study. Additionally, it will boost the data volume. I suggest adding these sites to the analysis. There may be more, these were the main ones which sprang to mind. On a related note, Figure 1 can probably be simplified for clarity by using one symbol/color for all AERONET sites, and another for all CARSNET sites. Splitting by year isn't necessary, in my view, and just complicates things.

Response: The AERONET sites are added, including those in HongKong.

11. The title of the manuscript suggests a broader scope than the analysis, since the analysis only performs an inter-comparison in the context of AERONET/CARSNET

measurements. There are various other things which could be added, at least briefly. For example, climatologies of seasonal AOD from all three algorithms (from the 1 degree products), and maps showing the available data volume (e.g. number of days per season with data), since this is another feature which is important for many applications. Otherwise, the title should be amended to reflect the scope. However I would prefer that the analysis be extended because I think that this would be quite useful (and new, to my knowledge).

Response: The seasonal validation and analysis has been added, and we also take insight into more analysis to make the scope broader.

Please also note the supplement to this comment:
http://www.atmos-chem-phys-discuss.net/acp-2016-195/acp-2016-195-SC2-supplement.pdf

---

## Author Comment (AC1) · 9 May 2016

Dear Editor and Reviewers,

We highly appreciate the detailed valuable comments from the referees on our manuscript of "acp-2016-195". The suggestions are quite helpful and we have incorporated them in the new version of manuscript. We have referred to literatures and papers and re-analyzed the collected data and reconstructed the paper to improve the quality of our paper.

As below, I would like to clarify some of the points raised by the Reviewers. And we hope that the reviewers and the editors will be satisfied with our responses to the 'comments' and the revisions for the original manuscript.

[Figure]

Yours truly,

Yong Xue

In this study, authors validate three AATSR AOD products (ADV, ORAC and SU algorithm) provided by Aerosol_cci project over China in 2007, 2008 and 2010. It's been widely validated (compared with AERONET AOD) that these three algorithms have ability in retrieving AOD over land with high precision till 3/23/2016. However, the AERONET data has limitations as reference data that there were not enough AERONET sites built and the distribution of AERONET sites were unevenly in mainland China in 2007, 2008 and 2010 caused by large territory of mainland China. Authors introduce CARSNET data to be combined with AERONET data, making up for these limitations and improving reliability of reference data. On this basis, authors not only select common evaluation metrics, but also introduce new metrics, for example, the improved KAPPA coefficient as comprehensive evaluation metric, the DR for determination of AOD retrieved "outliers", the improved expected error envelope designed for characteristics of AATSR AOD products, etc. This study is a nice trial consisting many meaningful works and I would recommend publication if my following comments/suggestions can be adequately addressed.

Major comments: 1. The structure and composition of manuscript should follow the requests of official website of Atmospheric Chemistry and Physics (ACP). For example, keywords, team list, etc. should add to manuscript and team list exist in this manuscript.

Response: All required structure and composition will be added in new version of manuscript.

2. Figures in the manuscript should be clear and easily understood. The main method of this study is to validate three AATSR AOD products year by year for reason of different reference data available for authors. Readers could distinguish which sites in the Fig. 1 is from AERONET or CARSNET, but may not pick out the space distribution of ground-based data sites in same year easily. I recommend authors replot Fig. 1 of

"The distribution of selected AERONET&CARSNET sites in mainland China in 2007, 2008 and 2010", using one same color or type for sites available in one year.

Response: The figures have been replotted with clear and easy understood symbols and text, as Fig. 1 we have replotted the symbols of sites from different networks using different colors to make it clearer.

3. Also I suggest that the paper never use the word "good" to describe the results. The coefficient of correlation (CC) as one of main evaluation metric, which indicates whether there is any linear relationship among the points. Authors could not claim which performances of products is "good" or not "good" by any values of CC or other evaluation metrics. For example, when CC is high, the performances could be viewed as "good", when CC is low, the performance is also viewed "good". The word "good" may confuse readers, leading misunderstanding of conclusions in this study.

Response: The "good" or "well" terms have been replaced by quantitative description or comparative words. For example, in section 4 to section 6, we have refined our analysis using more detailed quantitative description to present readers easy understood analysis.

Specific comments:

Page 2 line 9, the influences of aerosol particles on cloud should cited the paper of Twomey published in 1974.

Response: This reference will be added.

Page 2 line 19, the word "because" should be replaced by other words like "including"

Response: This sentence has been revised.

Page 3 line 10, the word "more" should be removed

Response: This word has been removed.

Page 3 line 15-19, comparison of satellite retrievals with other high quality has limitations, could you illustrate it more clearly?

Response: We have added necessary illustration and cross-validation with MODIS C6 DT&SB merged datasets.

Page 3 lin26, is "Aerosol_CCI" or "Aerosol_cci" formal?

Response: The CCI official website uses "Aerosol_cci", therefore, we'll introduce "Aerosol_cci" in the following or revised paper.

Page 4 Tab. 1 the bottom row are same with header row, what's it useful? And in the title abbreviation "Tab." should avoid.

Response: Tab. 1 has been revised.

Page 9 Tab. 4 these statistics should be up to two decimal point.

Response: Relative statistics have been kept two places of decimal.

Page 9 line 3, this sentence has syntax error.

Response: This sentence has been revised in new version of manuscript.

Fig. 8 – Fig. 16, the places of titles should be same.

Response: The places of title in figures have been adjusted to the same.

Page 19 Tab. 5 these statistics should be up to two decimal points.

Response: Relative statistics have been kept two places of decimal as Tab. 4.

Page 24 line 29, in part of acknowledgements, the numbers of sites are inconsistent with mentioned as above.

Response: The numbers of ground-based sites have been corrected in new version of manuscript.

Please also note the supplement to this comment:
http://www.atmos-chem-phys-discuss.net/acp-2016-195/acp-2016-195-AC1-supplement.pdf
* * *
* * *
[Figure]

**Supplement:**

[revised manuscript text omitted]
. For the consistency of the metrics among different aerosol products, strong matches are determined using the expected error (EE) which shows the percent of retrievals falling within the expected error (EE) range. An EE envelope was introduced for retrieval of MODIS AOD (Kaufman et al., 1997; Chu et al., 2002) by means of sensitivity studies, as demonstrated by Eq. (1) and Eq. (2):

$$EE1 = \pm(0.05 + 0.15\tau) \tag{1}$$

$$EE2 = \pm(0.05 + 0.20\tau) \tag{2}$$

where $\tau$ represents the satellite-retrieved AOD. AATSR AOD retrievals are different from the MODIS AOD datasets. In this paper, we introduced the EE envelope based on the features of AOD underestimation and formation of the MODIS EE envelope, as demonstrated by Eq. (3) and Eq. (4):

$$EE3 = \pm(0.05 + f2 + (f1 + 0.15)\tau) \tag{3}$$

$$EE4 = \pm(0.05 + f2 + (f1 + 0.20)\tau) \tag{4}$$

where $f1$ is the slope of the regression line of scatter points and $f2$ is the correspondent intercept. In the process of retrieving AOD, underestimation tends to be caused by systematic error. Therefore, the EE envelopes suggested by Kaufman et al. or Chu et al. are not fit for validation of the AATSR AOD. In such EE design, i.e., with Eq. (3) and Eq. (4), the underestimation was taken into consideration by regarding the regression line as the centre, not the 1-1 line, for determining the accidental error.

Linear regression is the most basic and commonly used statistical method that allows us to summarize and study relationships between two quantitative variables. Pearson correlation coefficient (CC) measures the fraction of the total variability in the response that is accounted for by the retrieval and is only a measure of linear association between ground truth measurements and satellite retrieval values. Bias describes the average difference between satellite retrievals and ground AOD. Then, to determine how well the satellite data match the ground-based observation data, the relationship between them is explored. A regression equation and some basic statistics are shown on the scatter plot, including the correlation coefficient (CC) and root mean square error (RMSE):

$$CC = \frac{\sum_{i=1}^{n}(\tau_{aero,i} - \overline{\tau_{aero}})(\tau_{sat,i} - \overline{\tau_{sat}})}{\sqrt{\sum_{i=1}^{n}(\tau_{aero,i} - \overline{\tau_{aero}})^2}\sqrt{\sum_{i=1}^{n}(\tau_{sat,i} - \overline{\tau_{sat}})^2}} \tag{5}$$

$$RMSE = \sqrt{\frac{1}{n}\sum_{i=1}^{n}(\tau_{sat,i} - \tau_{aero,i})^2} \tag{6}$$

[revised manuscript text omitted]

**3.1 The ADV algorithm**

For 2007, the RMS error is 0.095, the lowest of all results, the CC is 0.885, and the distribution of collocated pairs in the scatter plot is concentrated near the regression, as shown in Fig. 2a. Most of the collocated pairs are within EE3 (approximately 74.5%), indicating that the satellite retrievals are consistent with AERONET/CARSNET data. The RMB is 0.704, and the regression line is $y = 0.77x - 0.02$, which reflects the tendency of underestimation. This type of underestimation is more severe with increasing AOD. Low dispersion and slight underestimation make the KAPPA coefficient high (0.473), demonstrating that the ADV algorithm performs well in calculating the AOD over China in 2007. The ADV algorithm is appropriate for the retrieval of low AODs, especially for those less than 1.0; thus, the MSA for 2007 is 0.244.

For 2008, the lower RMB (0.621) suggests more severe underestimation, and the lower CC (0.776) and higher RSE (0.130) indicate lower accuracy. Similar with 2007, the MSA of the ADV is 0.211. Therefore, the KAAPA coefficient, which measures the overall performance, is 0.329, lower than that of 2007. For 2010, the lowest RMS (0.089) and largest proportion of matches are located in EE3 (91.2%) and EE4 (93.4%), with the lowest accidental error of the three years. However, the KAPPA coefficient is 0.180, also the lowest of the three years.

The most obvious feature of the ADV algorithm is underestimation, as demonstrated in Fig. 2. The mean $\pm 2\sigma$ lines in the different ranges are almost within the EE4 lines for these three years. The highest MSA is 0.250 in 2007, and the lowest is 0.173 in 2010. The ADV algorithm can retrieve low AOD values with high accuracy. This "ability" is systematic for either

high AODs or low AODs. This also limits the range of application of the ADV algorithm, especially in calculating AODs in high value ranges.

[Figure]

**Fig. 2. Scatter plots of AATSR ADV L2 AOD products with ground-based data in China in 2007, 2008 and 2010. The dashed, dotted and blue solid lines represent the 1-1 line, EE4 line and regression line, respectively. The magenta points are means for specific ranges of AERO&CARS AOD, and the magenta lines are the mean $\pm\ 2\sigma$ for a certain range.**

**3.2 The ORAC algorithm**

The ORAC algorithm performed well for 2007, achieving a KAPPA coefficient of 0.474. However, the distribution of matches is dispersed in Fig. 3b, implying low CC (0.708) and high RMSE (0.206). In terms of the degree of fitness, its performance is not effective. However, there is no obvious trend of underestimation or overestimation, and the regression line is close to the 1-1 line. Only 50.4% of collocated pairs are within EE4, and most of the mean $\pm\ 2\sigma$ lines are out of the EE4 lines, suggesting that accidental errors influence the accuracy of the ORAC algorithm. The MSA of the ORAC is 0.324.

ORAC has the most matches of the three algorithms (see Fig. 3). Different from 2008, no obvious underestimation occurs in the results of 2007 and 2010, as demonstrated by the regression lines shown in Fig. 3b and 3c. For 2008, the RMB is 0.829, suggesting a slight underestimation trend. The applicability of ORAC is high, with MSA of 0.271. The collocated pairs are relatively dispersed, and almost all mean $\pm\ 2\sigma$ lines are out of EE4 lines, influencing the RMSE and CC. For 2010, the same dispersion of points in the scatter plot and low KAPPA coefficient are observed.

[Figure]

**Fig. 3. Scatter plots of AATSR ORAC L2 AOD products with ground-based data in China in 2007, 2008 and 2010.**

Overall, the ORAC algorithm tends to retrieve AODs unstably for either high AODs or low AODs and with slight underestimation in 2007. The results of 2008 and 2010 share common features, even though the regression lines are below the 1-1 lines, indicating that accidental error is larger than systematic error.

**3.3 The SU algorithm**

The SU algorithm performed well for all three years, achieving KAPPA coefficients of 0.409, 0.484 and 0.520, respectively. Large proportions of matches are within EE3 and EE4, and almost all of the mean $\pm$ 2$\sigma$ lines are within the EE4 lines, both suggesting that the matches are concentrated in small regions around the regression line. The RMBs are 0.816, 0.713 and 0.720 for 2007, 2008 and 2010, respectively, demonstrating the underestimation of the SU product. The applicability of SU is high, with MSA of 0.293 for 2008.

The most obvious feature of the SU algorithm is its stability in retrieving AOD for different years or different regions (Fig. 4). The MSA ranges from 0.270 for 2010 to 0.330 for 2007, and the KAPPA coefficient is from 0.520 to 0.409, which suggests that the SU algorithm performed better in retrieving low AODs. The SU algorithm has the best performance in terms of AOD retrieval, as it has the highest KAPPA coefficient (0.520).

[Figure]

**Fig. 4. Scatter plots of AATSR SU L2 AOD products with ground-based data in China in 2007, 2008 and 2010.**

Overall, the SU algorithm can be applied to retrieve AOD in different ranges with high precision. Factors influencing the performance of the SU algorithm include small systematic error and even smaller accidental error.

**4. Uncertainty analysis based on aerosol loading**

5   In the previous section, we validated all three AOD products over mainland China in 2007, 2008 and 2010, discovering that all three products tend to exhibit underestimation to some extent. For the purpose of ascertaining the causes of the underestimation, in this section, we focus on analysing the AOD uncertainties leading to differences between retrieved AODs and ground-based AODs in special conditions. Collocated pairs are divided into three groups according to aerosol loading, including light loading ($\tau < 0.15$), heavy loading ($\tau > 0,4$), and moderate loading (Levy et al., 2010). It is obvious

10   that the AOD bias increases with increasing AOD for all three products. These products have one feature in common, that is, the AOD bias tends to be negative, which indicates that the underestimation becomes more significant with increasing aerosol loading. The ADV and SU algorithms perform well in estimating AOD, i.e., with little underestimation, when aerosol loading is low (light loading) (Fig. 5).

[Figure]

**Fig. 5. Scatter plot of AERONET&CARSNET AODs with ADV AOD bias or uncertainties, and histogram of AOD bias. Colours represent different groups: blue denotes light loading, green denotes moderate loading, and red denotes heavy loading. Basic**

**statistics are displayed in the top left corner, including the number of scattered points, MBE and linear regression equation (Fit). The text on the bottom with different colours describes the basic statistics of each group. Each group has one box, the bottom and top borders of which represent MBE + 2σ and MBE − 2σ, respectively, containing 96% of scattered points from each group. The centre line of each box represents the MBE of each group. The blue line is the regression line of all scattered points.**

5    Under complex conditions, the ORAC overestimates AOD in regions of light loading and moderate loading compared with the AEROENT, as shown in Fig. 6a-6b. Compared to the CARSNET data, overestimation occurs for light loading, and this overestimation is mainly due to two points with large error. In the moderate loading region, the MBE tends to be positive in Fig. 6a, probably because the distribution of AEROENT sites is uneven, as most sites are located in eastern China.

The top and bottom borders of the box we draw represent the interval of $[-2\sigma, 2\sigma]$, which contains most of the data
10   (approximately 95%) for a given group. The data outside the box are "possible outliers" based on the largest error contained in each group. Those "possible outliers" have one feature in common in that the corresponding points in the bias scatter plot are far away from other points. Otherwise, the points below or above the box are different. If a point is above the box, which indicates that the satellite-retrieved AOD is larger than the ground-based observed AOD, this "outlier" tends to be caused by a residual cloud. The ground-based network measures AOD from only one point; however, the satellite-retrieved AODs in
15   each collocated pairs are an average of 25 pixels. Any one of these 25 pixels with a cloud residual will lead to an increased AOD in a collocated pair. Therefore, we conclude that the "outliers" above the box are possibly caused by a cloud residual. From this view, one point above the box corresponds to each aerosol loading for the ADV product. The "outliers" are concentrated in the light loading region and moderate loading region for the SU product (Fig. 7). The situation of the ORAC is relatively complex; "outliers" occur in the light loading region, which makes the box of the light loading much larger than
20   that of the moderate loading region in 2007 and 2010 (Fig. 6a and 6c).

[revised manuscript text omitted]

---

## Author Comment (AC2) · 9 May 2016

Dear Editor and Reviewers,

We highly appreciate the detailed valuable comments from the referees on our manuscript of "acp-2016-195". The suggestions are quite helpful and we have incorporated them in the revised paper. We have referred to literatures and papers and re-analyzed the collected data and reconstructed the paper to improve the quality of our paper.

As below, I would like to clarify some of the points raised by the Reviewers. And we hope that the reviewers and the editors will be satisfied with our responses to the 'comments' and the revisions for the original manuscript.

[Figure]

Yours truly,

Yong Xue
 This paper validates 3 algorithms (SU, ADV, ORAC) for determining AOD from the European AATSR sensor against Sun photometer data in China (from AERONET and CARSNET). The topic is relevant to ACP. The work is important because these European products have not been as well-known as NASA ones, and have undergone a lot of development in the European CCI projects, so it is a good time to do some more thorough validation of these data sets. This is especially true for China since the aerosol loading is high and variable, and CARSNET has monitoring stations in some areas where AERONET is lacking.

have read through the manuscript several times and, while it is promising, there are some things which are unclear/invalid or I think not useful, and some important things which should be added to make the analysis more complete and useful. The phrasing is odd in some places and there are a number of typos (e.g. AEROENT in some places instead of AERONET) so I think that the manuscript will need some copy editing by the production office. I appreciate that English is not the authors' first language and the writing is not bad, it is just unclear in some cases. I therefore recommend some content revisions, to address the points below. I would like to review a revised version and think that another round of reviews will be necessary because the structure/content of the manuscript might change a lot. Here are my main points:

Response: The English of the manuscript has been edited by the Elsevier's Language Services.

1. Abstract: some of the sentences could probably be shortened (e.g. first and second

can be combined, as can third and fourth).

Response: The sentences have been shortened and refined in new version of abstract.

2. Introduction and start of section 2: It would be good to add a bit more information about the AATSR sensor here, like launch/end dates, swath width. A brief discussion of the differences between the algorithms should be included, to help understand why they give different results. From the analysis, the performance and the spatial coverage are both different between the three algorithms, so some insight into what in the algorithms is responsible would be welcome.

Response: More details and information about the AATSR instrument and retrieval algorithms have been added, furthermore, a brief analysis and discussion of the differences between the retrieval algorithms have also been added in the revised version of manuscript. These information will help readers to have a deep insight about the differences of validation result we have made in this study.

3. Statistics. Some of the metrics presented here are questionable in their relevance and I think that there are simpler and clearer alternatives. Specifically, the EE envelopes quoted here for Equations 1, 2 are for the MODIS instrument, not AATSR. AATSR is quite different (two views, fewer wavelengths) so there is no reason to expect that an AOD retrieval for AATSR would have the same type of behavior. One might expect that the error formulation would be closer to that of MISR. Further, Equations 3 and 4 are basically expressing a confidence envelope around a regression line. This is not really useful since it is just the noise around the relationship and not so dependent on the actual error in the retrievals. So comparing this between algorithms does not really make sense. A well-correlated but very biased retrieval would appear 'better' by this metric than a poorer-correlated but unbiased one, while for an actual scientific application, the unbiased one may in some cases be more useful. Further, least-squares linear regression is invalid for AOD retrievals because aerosol data violate the assumptions of this technique (see e.g. http://people.duke.edu/âĹijrnau/testing.htm ; the AOD

data validate assumptions 3 and 4 that linear least-squares regression makes, possibly 1 and 2 as well, and as a result the results obtained are not statistically valid). I know that a lot of people do least-squares linear regression because it is easy, but it is still wrong for this application. So, a better alternative is just to present statistics of bias and RMS error as a function of AOD, similar to what is shown in e.g. Figure 5 and the magenta bins in Figure 2. So I suggest that the EE discussion here and linear regressions be discarded entirely, and more prominence should be given to statistics subset into different regimes (e.g. low AOD, moderate AOD, high AOD; perhaps also splits based on Angstrom exponent for the high-AOD regimeïijĽ, as retrieval errors are often type-dependent as well). The kappa coefficient is probably fine. So, accounting for this comment would somewhat streamline and improve sections 2 and 3. Finally, presenting statistics to 3 significant figures is overkill and paints a picture of them being more robust than they probably are; 2 significant figures is probably enough.

Response: One objective of this manuscript is to evaluate different statistical metric for the validation of quantitative remote sensing. Different statistical metric shows different meaning and is used for different purpose. Linear regression is the most basic and commonly used statistical method that allows us to summarize and study relationships between two quantitative variables. Pearson correlation coefficient (CC) measures the fraction of the total variability in the response that is accounted for by the retrieval and is only a measure of linear association between ground truth measurements and satellite retrieval values. Bias describes the average difference between satellite retrievals and ground AOD. For the consistency of the metric among different aerosol products, it is better to show the percent of retrievals falling within the expected error (EE) range.

4. Retrieval errors. As I understand it, the CCI project means that the data products also provide their own estimates of the uncertainty in the AOD retrieval for every pixel. This is an important point, since pixel-level uncertainties are very useful for many applications. However, it is not discussed in the paper. How do these uncertainty estimates compare to the retrieval errors observed?

Response: The satellite retrieved AOD in each collocated pair are means of retrievals in 5 × 5 sampling frame. On this basis, we calculate means of uncertainty estimates in sampling area for each collocated pair as sizes of circles in scatter plot. In section 4 and 5, we reanalyze validation results of different algorithms, including comparison of uncertainty estimates and retrievals error observed.

5. Figures 2-4 and discussion in Section 3. I don't see any advantage to splitting out the points by year. It would be easier to combine all points from one algorithm into one panel, not 3. This would also let you combine Figures 2-4 into one figure for a side-by-side comparison of the three algorithms. Also, as discussed before, I would delete the regression and EE lines here since they are not very meaningful. The magenta symbols and lines for the binned data are enough here. Also, the color scales used in these figures are not mentioned and can probably be removed (either show individual points without a color scale or a density plot with a color scale). I also don't see any good reason to split the discussion of statistics out by year either. The data volume is not very large, so year to year differences are probably resulting from sampling and not statistically meaningful. Looking at the bigger picture of all data together is more statistically robust and gives a clearer picture. I don't believe any insight is gained by splitting the analysis up year by year.

Response: All points from one algorithm have been combined to make results more statistically robust and remove unnecessary plots. The colors of points in new scatter plot represent standard deviation of retrievals in sampling area for the purpose of finding influence on retrieving performance of sampling. We also keep the comparisons for each year as we would like to see the differences for each year. We added one section on the analysis of seasonal behaves of three algorithms.

6. Figures 5-7: Similar to the last comment: are the different panels the different years? It doesn't say anywhere but I infer that is the case. Again, these figures could be streamlined into one because clearly the biases are similar between years, this will be more robust, and will allow for a more direct comparison of the 3 algorithms.

Additionally, I don't think the histograms (bottom panels) here are useful since they don't provide any information which is not seen clearly in the top panels, so these could be deleted. Also, for the same reason as before, the linear regressions are invalid and should be deleted, just showing the binned values is enough.

Response: New scatter plots have been made, combining all points from one algorithm.

7. A similar plot to the bias plots could be created for RMSE. This would be a clearer way to show and compare the AOD-dependence of the retrieval error than the EE3/EE4 metrics.

Response: RMS error has been added in plot and statistic table.

8. In the discussion of the results, a lot of the time terms like "good" and "well" are used to describe performance. These are "weasel words" and should be avoided. What is "good" is only really relevant relative to a specific application (e.g. good enough to do X) or compared to the state of the art. I suggest rewording to avoid these words and be more quantitative where possible, or else stick to comparative terms (e.g. say when the data sets are similar to or better than each other). Also, some discussion of results compared to validation of other sensors (the main ones being MODIS/MISR) could be included, as these all have published validation for their aerosol products, and this would give a sense of how the AATSR data perform relative to the other available data products. Right now the paper more or less reads like AATSR is the only satellite option.

Response: The "weasel words" like "good" or "well" have been replaced by details of RMSE and KAPPA coefficient or comparative words. We compare and analyze AATSR AOD with "Deep Blue" and "Dark Target" 10km×10km AOD data from MODIS Collection 6 datasets which has been widely validated.

9. Do the retrievals provide other information like Angstrom exponent? From other

references, I believe so. This quantity is commonly compared with AERONET measurements, so it should be easy to extend the analysis to look at this as well using the same basic approaches. This might provide more insight for the differences between the data sets, if the algorithms are making very different assumptions about what sort of aerosol is present. This would help overcome one of the weaknesses of the paper, i.e. that the comparison is presented without any sort of discussion about why the three data sets are different and how to improve them (which would be very useful information).

Response: The CARSNET dataset provides AOD and angstrom exponent (440-870) only, otherwise the ADV provides angstrom exponent (550-670) only, ORAC provides angstrom exponent (550-870) only and SU provides angstrom exponent (550-870) only. Comparison between these data may be invalid.

10. There are at least two more AERONET sites in China which provided data in the study period, but which were not used in the analysis. These are both in Hong Kong: Hong_Kong_Hok_Tsui and Hong_Kong_PolyU. Why were these not used? If the objective (as stated) is to provide coverage over broad areas of China, then it would make sense to include them, since the data are freely available and there are no other sites used in this part of China. These sites are very close to the coast so also provide an additional type of environment to analyze, compared to the other sites presently included in the study. Additionally, it will boost the data volume. I suggest adding these sites to the analysis. There may be more, these were the main ones which sprang to mind. On a related note, Figure 1 can probably be simplified for clarity by using one symbol/color for all AERONET sites, and another for all CARSNET sites. Splitting by year isn't necessary, in my view, and just complicates things.

Response: The AERONET sites are added, including those in HongKong.

11. The title of the manuscript suggests a broader scope than the analysis, since the analysis only performs an inter-comparison in the context of AERONET/CARSNET

measurements. There are various other things which could be added, at least briefly. For example, climatologies of seasonal AOD from all three algorithms (from the 1 degree products), and maps showing the available data volume (e.g. number of days per season with data), since this is another feature which is important for many applications. Otherwise, the title should be amended to reflect the scope. However I would prefer that the analysis be extended because I think that this would be quite useful (and new, to my knowledge).

Response: The seasonal validation and analysis has been added, and we also take insight into more analysis to make the scope broader.

Please also note the supplement to this comment:
http://www.atmos-chem-phys-discuss.net/acp-2016-195/acp-2016-195-AC2-supplement.pdf

**Supplement:**

[revised manuscript text omitted]
. For the consistency of the metrics among different aerosol products, strong matches are determined using the expected error (EE) which shows the percent of retrievals falling within the expected error (EE) range. An EE envelope was introduced for retrieval of MODIS AOD (Kaufman et al., 1997; Chu et al., 2002) by means of sensitivity studies, as demonstrated by Eq. (1) and Eq. (2):

$$EE1 = \pm(0.05 + 0.15\tau) \tag{1}$$

$$EE2 = \pm(0.05 + 0.20\tau) \tag{2}$$

where $\tau$ represents the satellite-retrieved AOD. AATSR AOD retrievals are different from the MODIS AOD datasets. In this paper, we introduced the EE envelope based on the features of AOD underestimation and formation of the MODIS EE envelope, as demonstrated by Eq. (3) and Eq. (4):

$$EE3 = \pm(0.05 + f2 + (f1 + 0.15)\tau) \tag{3}$$

$$EE4 = \pm(0.05 + f2 + (f1 + 0.20)\tau) \tag{4}$$

where $f1$ is the slope of the regression line of scatter points and $f2$ is the correspondent intercept. In the process of retrieving AOD, underestimation tends to be caused by systematic error. Therefore, the EE envelopes suggested by Kaufman et al. or Chu et al. are not fit for validation of the AATSR AOD. In such EE design, i.e., with Eq. (3) and Eq. (4), the underestimation was taken into consideration by regarding the regression line as the centre, not the 1-1 line, for determining the accidental error.

Linear regression is the most basic and commonly used statistical method that allows us to summarize and study relationships between two quantitative variables. Pearson correlation coefficient (CC) measures the fraction of the total variability in the response that is accounted for by the retrieval and is only a measure of linear association between ground truth measurements and satellite retrieval values. Bias describes the average difference between satellite retrievals and ground AOD. Then, to determine how well the satellite data match the ground-based observation data, the relationship between them is explored. A regression equation and some basic statistics are shown on the scatter plot, including the correlation coefficient (CC) and root mean square error (RMSE):

$$CC = \frac{\sum_{i=1}^{n}(\tau_{aero,i} - \overline{\tau_{aero}})(\tau_{sat,i} - \overline{\tau_{sat}})}{\sqrt{\sum_{i=1}^{n}(\tau_{aero,i} - \overline{\tau_{aero}})^2}\sqrt{\sum_{i=1}^{n}(\tau_{sat,i} - \overline{\tau_{sat}})^2}} \tag{5}$$

$$RMSE = \sqrt{\frac{1}{n}\sum_{i=1}^{n}(\tau_{sat,i} - \tau_{aero,i})^2} \tag{6}$$

[revised manuscript text omitted]

**3.1 The ADV algorithm**

For 2007, the RMS error is 0.095, the lowest of all results, the CC is 0.885, and the distribution of collocated pairs in the scatter plot is concentrated near the regression, as shown in Fig. 2a. Most of the collocated pairs are within EE3 (approximately 74.5%), indicating that the satellite retrievals are consistent with AERONET/CARSNET data. The RMB is 0.704, and the regression line is $y = 0.77x - 0.02$, which reflects the tendency of underestimation. This type of underestimation is more severe with increasing AOD. Low dispersion and slight underestimation make the KAPPA coefficient high (0.473), demonstrating that the ADV algorithm performs well in calculating the AOD over China in 2007. The ADV algorithm is appropriate for the retrieval of low AODs, especially for those less than 1.0; thus, the MSA for 2007 is 0.244.

For 2008, the lower RMB (0.621) suggests more severe underestimation, and the lower CC (0.776) and higher RSE (0.130) indicate lower accuracy. Similar with 2007, the MSA of the ADV is 0.211. Therefore, the KAAPA coefficient, which measures the overall performance, is 0.329, lower than that of 2007. For 2010, the lowest RMS (0.089) and largest proportion of matches are located in EE3 (91.2%) and EE4 (93.4%), with the lowest accidental error of the three years. However, the KAPPA coefficient is 0.180, also the lowest of the three years.

The most obvious feature of the ADV algorithm is underestimation, as demonstrated in Fig. 2. The mean $\pm 2\sigma$ lines in the different ranges are almost within the EE4 lines for these three years. The highest MSA is 0.250 in 2007, and the lowest is 0.173 in 2010. The ADV algorithm can retrieve low AOD values with high accuracy. This "ability" is systematic for either

high AODs or low AODs. This also limits the range of application of the ADV algorithm, especially in calculating AODs in high value ranges.

[Figure]

**Fig. 2. Scatter plots of AATSR ADV L2 AOD products with ground-based data in China in 2007, 2008 and 2010. The dashed, dotted and blue solid lines represent the 1-1 line, EE4 line and regression line, respectively. The magenta points are means for specific ranges of AERO&CARS AOD, and the magenta lines are the mean $\pm\ 2\sigma$ for a certain range.**

**3.2 The ORAC algorithm**

The ORAC algorithm performed well for 2007, achieving a KAPPA coefficient of 0.474. However, the distribution of matches is dispersed in Fig. 3b, implying low CC (0.708) and high RMSE (0.206). In terms of the degree of fitness, its performance is not effective. However, there is no obvious trend of underestimation or overestimation, and the regression line is close to the 1-1 line. Only 50.4% of collocated pairs are within EE4, and most of the mean $\pm\ 2\sigma$ lines are out of the EE4 lines, suggesting that accidental errors influence the accuracy of the ORAC algorithm. The MSA of the ORAC is 0.324.

ORAC has the most matches of the three algorithms (see Fig. 3). Different from 2008, no obvious underestimation occurs in the results of 2007 and 2010, as demonstrated by the regression lines shown in Fig. 3b and 3c. For 2008, the RMB is 0.829, suggesting a slight underestimation trend. The applicability of ORAC is high, with MSA of 0.271. The collocated pairs are relatively dispersed, and almost all mean $\pm\ 2\sigma$ lines are out of EE4 lines, influencing the RMSE and CC. For 2010, the same dispersion of points in the scatter plot and low KAPPA coefficient are observed.

[Figure]

**Fig. 3. Scatter plots of AATSR ORAC L2 AOD products with ground-based data in China in 2007, 2008 and 2010.**

Overall, the ORAC algorithm tends to retrieve AODs unstably for either high AODs or low AODs and with slight underestimation in 2007. The results of 2008 and 2010 share common features, even though the regression lines are below the 1-1 lines, indicating that accidental error is larger than systematic error.

**3.3 The SU algorithm**

The SU algorithm performed well for all three years, achieving KAPPA coefficients of 0.409, 0.484 and 0.520, respectively. Large proportions of matches are within EE3 and EE4, and almost all of the mean $\pm$ 2$\sigma$ lines are within the EE4 lines, both suggesting that the matches are concentrated in small regions around the regression line. The RMBs are 0.816, 0.713 and 0.720 for 2007, 2008 and 2010, respectively, demonstrating the underestimation of the SU product. The applicability of SU is high, with MSA of 0.293 for 2008.

The most obvious feature of the SU algorithm is its stability in retrieving AOD for different years or different regions (Fig. 4). The MSA ranges from 0.270 for 2010 to 0.330 for 2007, and the KAPPA coefficient is from 0.520 to 0.409, which suggests that the SU algorithm performed better in retrieving low AODs. The SU algorithm has the best performance in terms of AOD retrieval, as it has the highest KAPPA coefficient (0.520).

[Figure]

**Fig. 4. Scatter plots of AATSR SU L2 AOD products with ground-based data in China in 2007, 2008 and 2010.**

Overall, the SU algorithm can be applied to retrieve AOD in different ranges with high precision. Factors influencing the performance of the SU algorithm include small systematic error and even smaller accidental error.

**4. Uncertainty analysis based on aerosol loading**

5   In the previous section, we validated all three AOD products over mainland China in 2007, 2008 and 2010, discovering that all three products tend to exhibit underestimation to some extent. For the purpose of ascertaining the causes of the underestimation, in this section, we focus on analysing the AOD uncertainties leading to differences between retrieved AODs and ground-based AODs in special conditions. Collocated pairs are divided into three groups according to aerosol loading, including light loading ($\tau < 0.15$), heavy loading ($\tau > 0,4$), and moderate loading (Levy et al., 2010). It is obvious

10   that the AOD bias increases with increasing AOD for all three products. These products have one feature in common, that is, the AOD bias tends to be negative, which indicates that the underestimation becomes more significant with increasing aerosol loading. The ADV and SU algorithms perform well in estimating AOD, i.e., with little underestimation, when aerosol loading is low (light loading) (Fig. 5).

[Figure]

**Fig. 5. Scatter plot of AERONET&CARSNET AODs with ADV AOD bias or uncertainties, and histogram of AOD bias. Colours represent different groups: blue denotes light loading, green denotes moderate loading, and red denotes heavy loading. Basic**

**statistics are displayed in the top left corner, including the number of scattered points, MBE and linear regression equation (Fit). The text on the bottom with different colours describes the basic statistics of each group. Each group has one box, the bottom and top borders of which represent MBE + 2σ and MBE − 2σ, respectively, containing 96% of scattered points from each group. The centre line of each box represents the MBE of each group. The blue line is the regression line of all scattered points.**

5    Under complex conditions, the ORAC overestimates AOD in regions of light loading and moderate loading compared with the AEROENT, as shown in Fig. 6a-6b. Compared to the CARSNET data, overestimation occurs for light loading, and this overestimation is mainly due to two points with large error. In the moderate loading region, the MBE tends to be positive in Fig. 6a, probably because the distribution of AEROENT sites is uneven, as most sites are located in eastern China.

The top and bottom borders of the box we draw represent the interval of $[-2\sigma, 2\sigma]$, which contains most of the data
10   (approximately 95%) for a given group. The data outside the box are "possible outliers" based on the largest error contained in each group. Those "possible outliers" have one feature in common in that the corresponding points in the bias scatter plot are far away from other points. Otherwise, the points below or above the box are different. If a point is above the box, which indicates that the satellite-retrieved AOD is larger than the ground-based observed AOD, this "outlier" tends to be caused by a residual cloud. The ground-based network measures AOD from only one point; however, the satellite-retrieved AODs in
15   each collocated pairs are an average of 25 pixels. Any one of these 25 pixels with a cloud residual will lead to an increased AOD in a collocated pair. Therefore, we conclude that the "outliers" above the box are possibly caused by a cloud residual. From this view, one point above the box corresponds to each aerosol loading for the ADV product. The "outliers" are concentrated in the light loading region and moderate loading region for the SU product (Fig. 7). The situation of the ORAC is relatively complex; "outliers" occur in the light loading region, which makes the box of the light loading much larger than
20   that of the moderate loading region in 2007 and 2010 (Fig. 6a and 6c).

[revised manuscript text omitted]

---

## Author Response (AR1)

Dear Editor and Reviewers,

We highly appreciate the detailed valuable comments from the referees on our manuscript of "acp-2016-195". The suggestions are quite helpful and we have incorporated them in the new version of manuscript. We have referred to literatures and papers and re-analyzed the collected data and reconstructed the paper to improve the quality of our paper.

As below, I would like to clarify some of the points raised by the Reviewers. And we hope that the reviewers and the editors will be satisfied with our responses to the 'comments' and the revisions for the original manuscript.

Thanks so much.

Yours Sincerely,

Yong Xue

In this study, authors validate three AATSR AOD products (ADV, ORAC and SU algorithm) provided by Aerosol_cci project over China in 2007, 2008 and 2010. It's been widely validated (compared with AERONET AOD) that these three algorithms have ability in retrieving AOD over land with high precision till 3/23/2016. However, the AERONET data has limitations as reference data that there were not enough AERONET sites built and the distribution of AERONET sites were unevenly in mainland China in 2007, 2008 and 2010 caused by large territory of mainland China. Authors introduce CARSNET data to be combined with AERONET data, making up for these limitations and improving reliability of reference data. On this basis, authors not only select common evaluation metrics, but also introduce new metrics, for example, the improved KAPPA coefficient as comprehensive evaluation metric, the DR for determination of AOD retrieved "outliers", the improved expected error envelope designed for characteristics of AATSR AOD products, etc. This study is a nice trial consisting many meaningful works and I would recommend publication if my following comments/suggestions can be adequately addressed.

Many thanks for your positive comments.

Major comments:
1. The structure and composition of manuscript should follow the requests of official website of Atmospheric Chemistry and Physics (ACP). For example, keywords, team list, etc. should add to manuscript and team list exist in this manuscript.

Response: All required structure and composition have been added in revised version of manuscript.

2. Figures in the manuscript should be clear and easily understood. The main method of this study is to validate three AATSR AOD products year by year for reason of different reference data available for authors. Readers could distinguish which sites in the Fig. 1 is from AERONET or CARSNET, but may not pick out the space distribution of ground-based data sites in same year easily. I recommend authors replot Fig. 1 of "The distribution of selected AERONET&CARSNET sites in mainland China in 2007, 2008 and 2010", using one same color or type for sites available in one year.

Response: The figures have been revised with clear and easy understood symbols and text, as Fig. 1 we have revised the symbols of sites from different networks using different colors to make it clearer.

3. Also I suggest that the paper never use the word "good" to describe the results. The coefficient of correlation (CC) as one of main evaluation metric, which indicates whether there is any linear relationship among the points. Authors could not claim which performances of products is "good" or not "good" by any values of CC or other evaluation metrics. For example, when CC is high, the performances could be viewed as "good", when CC is low, the performance is also viewed "good". The word "good" may confuse readers, leading misunderstanding of conclusions in this

study.

Response: The "good" or "well" terms have been replaced by quantitative description or comparative words. For example, in section 4 to section 6, we have refined our analysis using more detailed quantitative description to present readers easy understood analysis.

Specific comments:

Page 2 line 9, the influences of aerosol particles on cloud should cited the paper of Twomey published in 1974.
Response: This reference will be added.

Page 2 line 19, the word "because" should be replaced by other words like "including"
Response: This sentence has been revised.

Page 3 line 10, the word "more" should be removed
Response: This word has been removed.

Page 3 line 15-19, comparison of satellite retrievals with other high quality has limitations, could you illustrate it more clearly?
Response: We have added necessary illustration and cross-validation with MODIS C6 DT&SB merged datasets.

Page 3 lin26, is "Aerosol_CCI" or "Aerosol_cci" formal?
Response: The CCI official website uses "Aerosol_cci", therefore, we'll introduce "Aerosol_cci" in the following or revised paper.

Page 4 Tab. 1 the bottom row are same with header row, what's it useful? And in the title abbreviation "Tab." should avoid.
Response: Tab. 1 has been revised.

Page 9 Tab. 4 these statistics should be up to two decimal point.
Response: Relative statistics have been kept two places of decimal.

Page 9 line 3, this sentence has syntax error.
Response: This sentence has been revised in new version of manuscript.

Fig. 8 – Fig. 16, the places of titles should be same.
Response: The places of title in figures have been adjusted to the same.

Page 19 Tab. 5 these statistics should be up to two decimal points.
Response: Relative statistics have been kept two places of decimal as Tab. 4.

Page 24 line 29, in part of acknowledgements, the numbers of sites are inconsistent

with mentioned as above.

Response: The numbers of ground-based sites have been corrected in revised version of manuscript.

Dear Editor and Reviewers,

We highly appreciate the detailed valuable comments from the referees on our manuscript of "acp-2016-195". The suggestions are quite helpful and we have incorporated them in the revised paper. We have referred to literatures and papers and re-analyzed the collected data and reconstructed the paper to improve the quality of our paper.

As below, I would like to clarify some of the points raised by the Reviewers. And we hope that the reviewers and the editors will be satisfied with our responses to the 'comments' and the revisions for the original manuscript.

Yours truly,

Yong Xue
This paper validates 3 algorithms (SU, ADV, ORAC) for determining AOD from the European AATSR sensor against Sun photometer data in China (from AERONET and CARSNET). The topic is relevant to ACP. The work is important because these European products have not been as well-known as NASA ones, and have undergone a lot of development in the European CCI projects, so it is a good time to do some more thorough validation of these data sets. This is especially true for China since the aerosol loading is high and variable, and CARSNET has monitoring stations in some areas where AERONET is lacking.

have read through the manuscript several times and, while it is promising, there are some things which are unclear/invalid or I think not useful, and some important things which should be added to make the analysis more complete and useful. The phrasing is odd in some places and there are a number of typos (e.g. AEROENT in some places instead of AERONET) so I think that the manuscript will need some copy editing by the production office. I appreciate that English is not the authors' first language and the writing is not bad, it is just unclear in some cases. I therefore recommend some content revisions, to address the points below. I would like to review a revised version and think that another round of reviews will be necessary because the structure/content of the manuscript might change a lot. Here are my main points:

Response: The English of the manuscript has been edited by the Elsevier's Language Services.

1. Abstract: some of the sentences could probably be shortened (e.g. first and second can be combined, as can third and fourth).
   Response: The sentences have been shortened and refined in new version of abstract.

2. Introduction and start of section 2: It would be good to add a bit more information about the AATSR sensor here, like launch/end dates, swath width. A brief

discussion of the differences between the algorithms should be included, to help understand why they give different results. From the analysis, the performance and the spatial coverage are both different between the three algorithms, so some insight into what in the algorithms is responsible would be welcome.

Response: More details and information about the AATSR instrument and retrieval algorithms have been added, furthermore, a brief analysis and discussion of the differences between the retrieval algorithms have also been added in the revised version of manuscript. These information will help readers to have a deep insight about the differences of validation result we have made in this study.

3. Statistics. Some of the metrics presented here are questionable in their relevance and I think that there are simpler and clearer alternatives. Specifically, the EE envelopes quoted here for Equations 1, 2 are for the MODIS instrument, not AATSR. AATSR is quite different (two views, fewer wavelengths) so there is no reason to expect that an AOD retrieval for AATSR would have the same type of behavior. One might expect that the error formulation would be closer to that of MISR. Further, Equations 3 and 4 are basically expressing a confidence envelope around a regression line. This is not really useful since it is just the noise around the relationship and not so dependent on the actual error in the retrievals. So comparing this between algorithms does not really make sense. A well-correlated but very biased retrieval would appear 'better' by this metric than a poorer-correlated but unbiased one, while for an actual scientific application, the unbiased one may in some cases be more useful.

Further, least-squares linear regression is invalid for AOD retrievals because aerosol data violate the assumptions of this technique (see e.g. http://people.duke.edu/~rnau/testing.htm ; the AOD data validate assumptions 3 and 4 that linear least-squares regression makes, possibly 1 and 2 as well, and as a result the results obtained are not statistically valid). I know that a lot of people do least-squares linear regression because it is easy, but it is still wrong for this application.

So, a better alternative is just to present statistics of bias and RMS error as a function of AOD, similar to what is shown in e.g. Figure 5 and the magenta bins in Figure 2. So I suggest that the EE discussion here and linear regressions be discarded entirely, and more prominence should be given to statistics subset into different regimes (e.g. low AOD, moderate AOD, high AOD; perhaps also splits based on Angstrom exponent for the high-AOD regime), as retrieval errors are often type-dependent as well). The kappa coefficient is probably fine. So, accounting for this comment would somewhat streamline and improve sections 2 and 3. Finally, presenting statistics to 3 significant figures is overkill and paints a picture of them being more robust than they probably are; 2 significant figures is probably enough.

Response: One objective of this manuscript is to evaluate different statistical metric for the validation of quantitative remote sensing. Different statistical metric shows different meaning and is used for different purpose. Linear regression is the most basic and commonly used statistical method that allows us to summarize and study relationships between two quantitative variables. Pearson correlation coefficient (CC) measures the fraction of the total variability in the response that is

accounted for by the retrieval and is only a measure of linear association between ground truth measurements and satellite retrieval values. Bias describes the average difference between satellite retrievals and ground AOD. For the consistency of the metric among different aerosol products, it is better to show the percent of retrievals falling within the expected error (EE) range.

4. Retrieval errors. As I understand it, the CCI project means that the data products also provide their own estimates of the uncertainty in the AOD retrieval for every pixel. This is an important point, since pixel-level uncertainties are very useful for many applications. However, it is not discussed in the paper. How do these uncertainty estimates compare to the retrieval errors observed?

Response: The satellite retrieved AOD in each collocated pair are means of retrievals in $5 \times 5$ sampling frame. On this basis, we calculate means of uncertainty estimates in sampling area for each collocated pair as sizes of circles in scatter plot. In section 4 and 5, we reanalyze validation results of different algorithms, including comparison of uncertainty estimates and retrievals error observed.

5. Figures 2-4 and discussion in Section 3. I don't see any advantage to splitting out the points by year. It would be easier to combine all points from one algorithm into one panel, not 3. This would also let you combine Figures 2-4 into one figure for a side- by-side comparison of the three algorithms. Also, as discussed before, I would delete the regression and EE lines here since they are not very meaningful. The magenta symbols and lines for the binned data are enough here. Also, the color scales used in these figures are not mentioned and can probably be removed (either show individual points without a color scale or a density plot with a color scale).

I also don't see any good reason to split the discussion of statistics out by year either. The data volume is not very large, so year to year differences are probably resulting from sampling and not statistically meaningful. Looking at the bigger picture of all data together is more statistically robust and gives a clearer picture. I don't believe any insight is gained by splitting the analysis up year by year.

Response: All points from one algorithm have been combined to make results more statistically robust and remove unnecessary plots. The colors of points in new scatter plot represent standard deviation of retrievals in sampling area for the purpose of finding influence on retrieving performance of sampling. We also keep the comparisons for each year as we would like to see the differences for each year. We added one section on the analysis of seasonal behaves of three algorithms.

6. Figures 5-7: Similar to the last comment: are the different panels the different years? It doesn't say anywhere but I infer that is the case. Again, these figures could be streamlined into one because clearly the biases are similar between years, this will be more robust, and will allow for a more direct comparison of the 3 algorithms. Additionally, I don't think the histograms (bottom panels) here are useful since they don't provide any information which is not seen clearly in

the top panels, so these could be deleted. Also, for the same reason as before, the linear regressions are invalid and should be deleted, just showing the binned values is enough.

Response: New scatter plots have been made, combining all points from one algorithm.

7. A similar plot to the bias plots could be created for RMSE. This would be a clearer way to show and compare the AOD-dependence of the retrieval error than the EE3/EE4 metrics.

Response: RMS error has been added in plot and statistic table.

8. In the discussion of the results, a lot of the time terms like "good" and "well" are used to describe performance. These are "weasel words" and should be avoided. What is "good" is only really relevant relative to a specific application (e.g. good enough to do X) or compared to the state of the art. I suggest rewording to avoid these words and be more quantitative where possible, or else stick to comparative terms (e.g. say when the data sets are similar to or better than each other). Also, some discussion of results compared to validation of other sensors (the main ones being MODIS/MISR) could be included, as these all have published validation for their aerosol products, and this would give a sense of how the AATSR data perform relative to the other available data products. Right now the paper more or less reads like AATSR is the only satellite option.

Response: The "weasel words" like "good" or "well" have been replaced by details of RMSE and KAPPA coefficient or comparative words. We compare and analyze AATSR AOD with "Deep Blue" and "Dark Target" 10km×10km AOD data from MODIS Collection 6 datasets which has been widely validated.

9. Do the retrievals provide other information like Angstrom exponent? From other references, I believe so. This quantity is commonly compared with AERONET measurements, so it should be easy to extend the analysis to look at this as well using the same basic approaches. This might provide more insight for the differences between the data sets, if the algorithms are making very different assumptions about what sort of aerosol is present. This would help overcome one of the weaknesses of the paper, i.e. that the comparison is presented without any sort of discussion about why the three data sets are different and how to improve them (which would be very useful information).

Response: The CARSNET dataset provides AOD and angstrom exponent (440-870) only, otherwise the ADV provides angstrom exponent (550-670) only, ORAC provides angstrom exponent (550-870) only and SU provides angstrom exponent (550-870) only. Comparison between these data may be invalid.

10. There are at least two more AERONET sites in China which provided data in the study period, but which were not used in the analysis. These are both

in Hong Kong: Hong_Kong_Hok_Tsui and Hong_Kong_PolyU. Why were these not used? If the objective (as stated) is to provide coverage over broad areas of China, then it would make sense to include them, since the data are freely available and there are no other sites used in this part of China. These sites are very close to the coast so also provide an additional type of environment to analyze, compared to the other sites presently included in the study. Additionally, it will boost the data volume. I suggest adding these sites to the analysis. There may be more, these were the main ones which sprang to mind. On a related note, Figure 1 can probably be simplified for clarity by using one symbol/color for all AERONET sites, and another for all CARSNET sites. Splitting by year isn't necessary, in my view, and just complicates things.

Response: The AERONET sites are added, including those in HongKong.

11.     The title of the manuscript suggests a broader scope than the analysis, since the analysis only performs an inter-comparison in the context of AERONET/CARSNET measurements. There are various other things which could be added, at least briefly. For example, climatologies of seasonal AOD from all three algorithms (from the 1 degree products), and maps showing the available data volume (e.g. number of days per season with data), since this is another feature which is important for many applications. Otherwise, the title should be amended to reflect the scope. However I would prefer that the analysis be extended because I think that this would be quite useful (and new, to my knowledge).

Response: The seasonal validation and analysis has been added, and we also take insight into more analysis to make the scope broader.

[revised manuscript text omitted]
 could can be transported a long distances to eastern China, and even to Japan (Takahashi, 2011), showing resulting in regional characteristicsdifferences.

**Tab. 1. Details of AATSR AOD products.**

| algorithm | version | sensor | Main parameters | Resolution coverage |
|---|---|---|---|---|
| ADV/ASV | 2.3 | AATSR | AOD,ANG | 10 km, 1° global |
| SU | 4.21 | AATSR | AOD,ANG | 10 km, 1° global |
| ORAC | 03.04 | AATSR | AOD, aerosol type | 10 km, 1° global |
| algorithm | version | sensor | Main parameters | Resolution coverage |
| ADV/ASV | 2.3 | AATSR | AOD,ANG | 10km,10 km, 1° global |
| SU | 4.21 | AATSR | AOD,ANG | 10km,10 km, 1° global |
| ORAC | 03.04 | AATSR | AOD, aerosol type | 10km,10 km, 1° global |
| algorithm | version | sensor | Main parameters | Resolution coverage |

[Figure]

AERONET&CARSNET sites in China

[Figure]

**Fig. 1. The distribution of selected AERONET&CARSNET sites in mainland China in 2007, 2008 and 2010. The blue and red points represent AERONET and CARNET sites, respectively.**

5   The measurements from another network, the CARSNET, equipped with calibrated CE-318 instruments, have the same accuracy as AERONET. The CARSNET has more sites than the  AERONET in mainland China, and the spatial distribution of the CARSNET sites  is distributed more evenly. Therefore, for the purpose of assessing different performances of these three AATSR L2 AOD products, we selected ground-based measurements from both of these two networks as reference data.

The AERONET provides AOD data at three data quality levels: Level 1.0 (unscreened), Level 1.5 (cloud-screened), and Level 2.0 (L2) (cloud screened and quality-assured) (http://aeronet.gsfc.nasa.gov/new_web/index.html). Here, we selected AERONET L2 data  that are screened and quality-assured. Because both  the AERO T and CARSNET data  are AATSR products without band-effective wavelength, we interpolated the ground-based data to  550-

5 nm wavelength. The AOD of the L2 datasets were compared with AEROENT&CARSNET observation data using scatter plots and linear-regression  of the data. The comparisons were made for  collocated satellite and ground-based observations (Ichoku et al., 2002), i.e. AOD pixels were selected within a spatial extent of +/−50 km  of ground-based station  and a time range of +/−30 min of the AATSR overpass from the ground-based measurements. At least 5 AATSR AOD retrievals and 2 AERONET/CARSNET observations are required in each collocation (Levy et al., 2010).

10 We  conducted collocations according to year (2007 2008 and 2010) and dataset (ADV ORAC and SU).  In total,  20 ground-based observation sites, including 12 AERONET sites and 8 CARSNET sites, were  in the Chinese territory in 2007, of which 6 AERONET and 8 CARSNET inland sites were selected. For 2008, we selected 8 AERONET and 24 CARSNET inland sites, for a total of 32 sites, ignoring the island sites and those near the shoreline. For 2010, only 6 CARSNET sites are available for us, and a total of 14 inland sites were selected with 8 AERONET inland sites (see Table 2).

15 **Table 2. Selected ground-based sites in China.**

|  | Network | inland | near shoreline | island | Total |
|---|---|---|---|---|---|
| 2007 | AERONET | 6 | 6 | 0 | 12 |
|  | CARSNET | 8 | 0 | 0 | 8 |
|  | Total | 14 | 6 | 0 | 20 |
| 2008 | AERONET | 8 | 7 | 0 | 15 |
|  | CARSNET | 24 | 1 | 0 | 25 |
|  | Total | 32 | 8 | 0 | 40 |
| 2010 | AERONET | 8 | 7 | 1 | 16 |
|  | CARSNET | 6 | 0 | 0 | 6 |
|  | Total | 14 | 7 | 1 | 20 |

**2.1 Statistics Metrics**

Collocated pairs are analyzed using statistical methods. For the consistency of the metrics among different aerosol products, strong matches are determined using the expected error (EE) which shows the percent of retrievals falling within the expected

20 error (EE) range. An EE envelope was  introduced for retrieval of MODIS AOD (Kaufman et al., 1997; Chu et al., 2002) by means of sensitivity studies, as demonstrated by Eq. (1) and Eq. (2):

$$EE1 = \pm(0.05 + 0.15\tau) \tag{1}$$

$$EE2 = \pm(0.05 + 0.20\tau) \tag{2}$$

where, τ represents the satellite-retrieved AOD. AATSR AOD retrievals are different from the MODIS AOD datasets., In this paper, we introduced the EE envelope based on the feature of AOD underestimation and formation of the MODIS EE envelope, as demonstrated by Eq. (3) and Eq. (4):

$$EE3 = \pm(0.05 + f2 + (f1 + 0.15)\tau) \tag{3}$$

$$EE4 = \pm(0.05 + f2 + (f1 + 0.20)\tau) \tag{4}$$

where, $f1$ is the slope of the regression line of scatter points and $f2$ is the correspondent intercept. In the process of retrieving AOD, underestimation tends to be caused by systematic error. Therefore, the EE envelopes suggested by Kaufman et al. or Chu et al. are not fit for validation of the AATSR AOD. In Such  EE design, i.e.,  with Eq. (3) and Eq. (4),  the underestimation was taken into consideration by, regarding the regression line as the cente, not the 1-1 line, for determining the accidental error.

Linear regression is the most basic and commonly used statistical method that allows us to summarize and study relationships between two quantitative variables. Pearson correlation coefficient (CC) measures the fraction of the total variability in the response that is accounted for by the retrieval and is only a measure of linear association between ground truth measurements and satellite retrieval values. 
[revised manuscript text omitted]
 the collocated pairs are within EE3 ( approximately 74.5%), indicating that the satellite retrievals  are consistent with AERONET/CARSNET data .

10 The RMB is 0.704, and the regression line is y = 0.77x – 0.02, which reflect  the tendency of underestimation. This  type of underestimation  is more severe  with increasing AOD . Low dispersion and slight underestimation make the KAPPA coefficient high (0.473),  demonstrating that the ADV algorithm  performs well in calculating the AOD over China in 2007. The ADV algorithm is appropriate for the retrieval of low AODs, especially for those less than 1.0,  thus, the MSA for 2007 is 0.244.

15 For 2008, the lower RMB (0.621)  suggests more severe underestimation, and the lower CC (0.776) and higher RSE (0.130)  indicate lower accuracy. Similar with 2007, the MSA of the ADV is 0.211. Therefore, the KAAPA coefficient, which  measures the overall performance, is 0.329, lower than  that of 2007. For 2010, the lowest RMS (0.089) and largest proportion of matches  d located in EE3 (91.2%) and EE4 (93.4%),  with the lowest  accidental error of the three years. However, the KAPPA coefficient is 0.180, also the lowest  of the three years.

The most obvious feature of the ADV algorithm is underestimation, as  demonstrated in Fig. 2. The mean ± 2σ lines in the different ranges are almost within the EE4 lines for these three years. The highest MSA is 0.250 in 2007, and the lowest  is 0.173 in 2010 . The ADV algorithm can retrieve low AOD values  with high accuracy. This "ability" is systematic for either high AODs or low AODs. This also limits the range of application of the ADV algorithm, especially in calculating AODs in high value ranges.

[Figure]

**Fig. 2. Scatter plots of AATSR ADV L2 AOD products with ground-based data in China in 2007, 2008 and 2010. The dashed, dotted and blue solid lines represent the 1-1 line, EE4 line and regression line, respectively. The magenta points are means for specific ranges of AERO&CARS AOD, and the magenta lines are the mean ± 2σ for a certain range.**

**3.2 The ORAC algorithm**

The ORAC algorithm  performed well in 2007, achieving a KAPPA coefficient of 0.474. However, the distribution of  matches is dispersed in Fig. 3b, implying low CC (0.708) and high RMSE (0.206). In terms of the degree of fitness, its performance is not effective. However, There 's no obvious trend of underestimation or overestimation, and the regression line is close to the 1-1 line. Only 50.4% of collocated pairs are within EE4, and most of the mean ± 2σ lines are out of the EE4 lines,  suggesting that accidental errors influence the accuracy of the ORAC algorithm. The MSA of the ORAC is 0.324.

 ORAC has the most matches of the three algorithms (see Fig. 3). Different from 2008,  obvious underestimation occurs in the results of 2007 and 2010, as demonstrated by the regression lines shown in Fig. 3b and 3c. For 2008, the RMB is 0.829,  suggesting a slight underestimation trend. The applicability of ORAC is high, with MSA of 0.271. The collocated pairs are relatively dispersed, and almost all mean ± 2σ lines are out of EE4 lines, influencing the RMSE and CC. For 2010, the same dispersion of points in the scattered plot and low KAPPA coefficient are observed.

[Figure]

**Fig. 3. Scatter plots of AATSR ORAC L2 AOD products with ground-based data in China in 2007, 2008 and 2010.**

Overall, the ORAC algorithm tends to retrieve AODs unstably for either high AODs or low AODs and with slight underestimation in 2007. The results of 2008 and 2010  share common features , even though the regression lines are below the 1-1 lines,  indicating that  accidental error  is larger than systematic error.

**3.3 The SU algorithm**

The SU algorithm  performed well for all three years,  achieving KAPPA coefficients of 0.409, 0.484 and 0.520, respectively. Large proportions of matches are within EE3 and EE4, and almost  all of the mean ± 2σ lines are within the EE4 lines, both  suggesting that the matches are concentrated in small regions around the regression line. The RMBs are 0.816, 0.713 and 0.720  for 2007, 2008 and 2010, respectively,  demonstrating the underestimation of the SU product. The applicability of SU is  high, with MSA of 0.293 for 2008.

The most obvious feature of the SU algorithm is its stability in retrieving AOD for different years or different regions (Fig. 4). The MSA  ranges from 0.270 for 2010 to 0.330 for 2007, and the KAPPA coefficient is from 0.520 to 0.409, which  suggests that the SU algorithm  performed better  in retrieving low AODs. The SU algorithm has the best performance in terms of AOD retrieval, as it has the highest KAPPA coefficient (0.520).

[Figure]

**Fig. 4. Scatter plots of AATSR SU L2 AOD products with ground-based data in China in 2007, 2008 and 2010.**

Overall, the SU algorithm  can be applied to retrieve AOD  in different range with high precision. Factors  influencing the performance of the SU algorithm include small systematic error and even smaller accidental error.

**4. Uncertainty analysis based on aerosol loading**

In the previous section, we  validated all three AOD products over mainland China in 2007, 2008 and 2010, discovering that all  three products tend to exhibit underestimation  to some extent. For the purpose of ascertaining the causes of the underestimation, in this section, we focus on analysing the AOD uncertainties leading to differences between retrieved AODs and ground-based AODs  in special conditions. Collocated pairs are divided into three groups according to aerosol loading, including light loading (τ < 0.15), heavy loading (τ > 0,4), and moderate loading (Levy et al., 2010). It is obvious that the AOD bias increases with increasing AOD for all three products. These products have one feature in common, that is, the AOD bias tends to be negative, which  indicates that the underestimation become more significant with increasing aerosol loading. The ADV and SU algorithms  perform well on estimating AOD , i.e., with little underestimation, when aerosol loading is low (light loading) (Fig. 5).

[Figure]

**Fig. 5.** Scatter plot of AERONET&CARSNET AODs with ADV AOD bias or uncertainties, and hist~r~ogram of AOD bias. Colo~u~rs represent different groups~:~, blue ~for~ denotes light loading, green ~for~ denotes moderate loading, and red ~for~ denotes heavy loading. Basic statistics are ~texted~ displayed ~on~ in the top left corner, including the number of scattered points, MBE and linear regression equa~ti~on (Fit). The text on the bottom with different colo~u~rs ~are~ describes the basic statistics of each group. Each group has one box, the bottom and top ~borders-bottom and borders-top~borders of which represent MBE + 2σ and MBE − 2σ, respectively, containing 96% ~of~ scattered points from each group. The cent~er~e line of each box represents the MBE of each group. The blue line is the regression line of all scattered points.

Under complex conditions, the ORAC overestimates AOD in regions of light loading and moderate loading compared with the AEROENT, as shown in Fig. 6a-6b. Compared to the CARSNET data, ~it also appears~ overestimation occurs for light loading, and this overestimation is ma~iea~inly due to two points with large error. In the moderate loading region, the MBE tends to be positive in Fig. 6a, probably because the distribution of AEROENT sites is uneven, ~that~ as most ~of~ sites are located in eastern China.

The top and bottom borders of the box we draw represent the interval of $[-2\sigma, 2\sigma]$, which contains most of the data (~about~ approximately 95%) for a given group. The data outside the box are "possible outliers" based on the~due to~ largest error contained in each group. Those "possible outliers" have one feature in common in that the corresponding points in the bias scatter plot are far away from other points. Otherwise, the points below or above the box are different. If a point~s are~ is above the box, which ~means~ indicates that the satellite-retrieved AOD ~are~ is larger than the ground-based observed AOD, ~those~ this "outlier~s~" tend~s~ to be caused by a residual cloud. ~Because~ The ground-based network measures AOD ~just~ from only one point~;:~ ~but~ however, the satellite--retrieved AODs in each collocated pairs are an average of 25 pixels. Any one of these 25 pixels with a cloud residual will lead to an increase ~of~d AOD in a collocated pair. Therefore, we ~make a conclusion~conclude that the "outliers" above the box are possibly caused by a cloud residual. From this view, ~there's~ one point above the box ~of~ corresponds to each aerosol loading ~respectively~ for the ADV product. ~This kind of~The "outliers" are concentrate~d~s in the~up~ light loading region and moderate loading region for the SU product (Fig. 7). The situation of the ORAC is relativel~y~ complex~:,~ ~it exists~ "outliers" occur in the light loading region, which makes the box of the light loading much larger than ~box~ that of the moderate loading region in 2007 and 2010 ~as shown in~(-Fig. 6a and 6c).

[Figure]

[Figure]

**Fig. 6.** Scatter plot of AERONET&CARSNET AODs with ORAC AOD bias or uncertainties and histrogram of AOD bias.

The points below the box are different from those above the box;, most of them are only below the box for due to heavy loading, indicating that the ability of estimating AOD will decreasedecreases with the increase ofincreasing aerosol loading, especially
5 in region ofthe heavy aerosol loading region.

[Figure]

**Fig. 7.** Scatter plot of AEROENT&CARSNET AODs with SU AOD bias or uncertainties and histrogram of AOD bias.

We make these groups because aerosols have exhibit different natures behaviours with different loading conditions. In general,
10 the bias or uncertainty of satellite-retrieved AOD will increase with the increase ofincreasing AOD or aerosol loading. As

[revised manuscript text omitted]

Xianghe is located at to the southeast of Beijing, having and has the same climatic pattern conditions as Beijing. About Approximately 98% of the surface is covered with urban land from according to the MCD12C1 data at extent of a 50 km × 50km 50 km area. The performances of these three algorithms are at the same high quality level with high quality. However, the ADV algorithm still underestimated AOD at a level of MBE = 0.12 in 2007 and 0.10 in 2008.

[Figure]

**Fig. 11. is the time Time series comparison of AATSR AOD with AERONET AOD over the site of at XiangHe in 2007, 2008 and 2010.**

Xilinhot is situated at 116.07°E, 43.95°N, 116.07°E, at the centre of the Xilinguole grassland. The main land cover is grassland (100%) from based on the MODIS MCD12C1 data, at with a spatial extent of 50 km × 50km. 50 km. The surface
10   circumstance and climate features of Xilinhot are much like similar to those of SACOL's, and the performances of the SU

algorithm  at these two sites is the same, i.e., both with high R and low RMSE . The ADV algorithm slightly underestimated AOD  with MBE of 0.10~0.13. The ORAC AOD showed  weak agreement with the Xilinhot data, mainly because  possible "outliers" exist in March to June 2008 and March 2010.

[Figure]

**Fig. 12. comparison of SU AOD with CARSNET AOD at Xilinhot in 2008 and 2010.**

**Table 5. Statistics of validation results of different products over different sites.**

| Site | Algorithm | N | MSA | MAA | MBE | MAE | RMSE | RMB | CC | KAPPA | Within EE3 | Within EE4 |
|---|---|---|---|---|---|---|---|---|---|---|---|---|
| | ADV | 33 | 0.346 | 0.462 | -0.116 | 0.122 | 0.088 | 0.748 | 0.916 | 0.341 | 84.9% | 90.9% |
| Linan | ORAC | 48 | 0.426 | 0.470 | -0.044 | 0.131 | 0.144 | 0.906 | 0.647 | 0.668 | 58.3% | 70.8% |
| | SU | 40 | 0.430 | 0.484 | -0.054 | 0.082 | 0.093 | 0.889 | 0.917 | 0.650 | 85.0% | 90.0% |
| | ADV | 46 | 0.156 | 0.285 | -0.129 | 0.132 | 0.068 | 0.547 | 0.763 | 0.283 | 89.1% | 91.3% |
| SACOL | ORAC | 74 | 0.286 | 0.314 | -0.028 | 0.102 | 0.170 | 0.910 | 0.683 | 0.595 | 67.6% | 73.0% |
| | SU | 49 | 0.265 | 0.291 | -0.027 | 0.062 | 0.072 | 0.908 | 0.849 | 0.878 | 77.6% | 83.7% |

| | | | | | | | | | | | | |
|---|---|---|---|---|---|---|---|---|---|---|---|---|
| | ADV | 52 | 0.172 | 0.297 | -0.125 | 0.131 | 0.087 | 0.578 | 0.780 | 0.339 | 78.9% | 90.4% |
| Shangdianzi | ORAC | 66 | 0.267 | 0.304 | -0.037 | 0.107 | 0.134 | 0.879 | 0.781 | 0.407 | 54.6% | 65.2% |
| | SU | 46 | 0.285 | 0.402 | -0.117 | 0.128 | 0.101 | 0.710 | 0.924 | 0.457 | 82.6% | 87.0% |
| | ADV | 33 | 0.184 | 0.284 | -0.100 | 0.102 | 0.070 | 0.649 | 0.921 | 0.169 | 97.0% | 97.0% |
| XiangHe | ORAC | 34 | 0.227 | 0.240 | -0.013 | 0.091 | 0.096 | 0.946 | 0.825 | 0.577 | 73.5% | 76.5% |
| | SU | 36 | 0.368 | 0.392 | -0.024 | 0.058 | 0.077 | 0.939 | 0.984 | 0.444 | 88.9% | 91.7% |
| | ADV | 49 | 0.082 | 0.198 | -0.116 | 0.117 | 0.046 | 0.414 | 0.814 | 0.148 | 95.9% | 95.9% |
| Xilinhot | ORAC | 110 | 0.190 | 0.182 | 0.008 | 0.109 | 0.166 | 1.043 | 0.517 | 0.389 | 42.7% | 48.2% |
| | SU | 61 | 0.140 | 0.220 | -0.081 | 0.085 | 0.063 | 0.634 | 0.937 | 0.444 | 88.9% | 91.7% |

For To guarantee of statistical reliability, there must be more than 30 collocated pairs in at one site. The determination of the surface cover at on each site is is according to based on the proportion (> 80% for one land type) of each land cover type from the MCD12C1 data at a spatial extent of 50 km × 50km. 50 km. If there's no one land cover type accounts for a's proportion

[revised manuscript text omitted]
, but however,whereas most are underestimated. Another two sites with dominateddominated by land cover of barren or sparsely vegetation vegetated land cover are Dunhang (about approximately 85%) and Tazhong (100%). The circumstance conditions in Tazhong is are complex, and there are is no obvious laws relationship between the CARSNET data and the ORAC AODs. Most of the DRs are less than 3, and a total of 8 DRs are larger than 3. It's basically identified that the oneThe DR in February is an "outlier", because the varying tendencies are different between the ORAC product and the ground-based data, and only this points wasindicative of overestimation.

**Table 6. DR dDistribution of DRs ofof specific sites.**

| Site | DR<1 | 1<DR<3 | 3<DR<5 | 5<DR | Total |
|---|---|---|---|---|---|
| Urumchi | 47 | 40 | 2 | 1 | 90 |
| Ejina | 51 | 43 | 1 | 1 | 96 |
| Tazhong | 63 | 17 | 5 | 3 | 88 |
| Dunhuang | 57 | 31 | 1 | 2 | 91 |

The ORAC product has the largest coverage at the cost expense of losing accuracy, especially exist ofin the presence of "outliers", and only the ORAC product has collocated collocate validation pairs over some sites at during each month in all three years. The ORAC algorithm underestimates AODs over Ejina, Tazhong and Dunhuang, but the "possible outliers" reduce the differences between the CARSNET data and the ORAC product. Xilinhot, Urumchi and SACOL have share the same main land cover of grasslands. The problem is that the underestimations over these sites are not at one the same level.

It is worth noting that the ORAC algorithm has the ability in calculating to calculate high AOD.; however, most of the AODs of whichhave DRs are larger than 3, indicating that the estimation of high AOD is unstable with and has large error, even to reduce the wholereducing the overall precision.

**6. Seasonal characteristics of three algorithms**

The mainland China, cross about 60 degree of longitude and 30 degree of latitude, is dominated by monsoon-driven climate. In such vast territory, there are big differences in climate pattern from western to eastern China. The main climate type in eastern and eastern coastal China is monsoon climate. For western China far from the ocean, the climate type is hybrid of monsoon and continental climate. In dry seasons (winter, first half of spring, and last half of autumn), poor vegetation coverage, loosen surface and winds in most northern China regions make coarse particles (sea salt and desert dust) into aerosol. Fine particles from coal combustion in winter and soot from straw burning in autumn is also important source of aerosol. In rainy seasons (mainly in summer), high vegetation blocks dust blowing into aerosol and reduce surface reflectance at visible wavelength. Table 7 shows the seasonal distribution of validation results of three algorithms. For the mainland China which is

located in Northern Hemisphere from 20°N to 55°N, the spring time starts from about March to May, the summer time starts from about June to August, the autumn time starts from about September to November and the winter time is about from December to February in next year.

5    **Table 7. Seasonal distribution of validation results of three algorithms.**

[revised manuscript text omitted]

---

## Author Response (AR3)

Dear Editor and Reviewers,

We highly appreciate the detailed valuable comments from the referees on our manuscript of "acp-2016-195". The suggestions are quite helpful and we have incorporated them in the revised paper. We have referred to PEEX scientific plan to improve the quality of our paper. We prefer to keep the analysis for both every year and all three years. But we have added a few lines following the reviewer's comments.

As below, I would like to clarify some of the points raised by the Reviewers. And we hope that the reviewers and the editors will be satisfied with our responses to the 'comments' and the revisions for the original manuscript. The supplement file is the manuscript with all markups.

Yours truly,

Yong Xue

Anonymous Referee #2

Suggestions for revision or reasons for rejection (will be published if the paper is accepted for final publication)
I reviewed two previous versions of this paper, and had concerns about the statistical techniques used and consequent validity and usefulness of the analysis. In this version the authors have improved by removing the problematic statistical analysis. I appreciate the authors' efforts on this, and recommend publication after a few minor corrections, as this will now be a useful intercomparison of three data sets. The manuscript will need some general proof-reading and copy-editing; my comments are below:

Page 3, line 22: I suggest changing 'cloud removal' to 'cloud screening' because 'removal' is often use to mean aerosols washed out of the atmosphere by clouds, while what the authors refer to is instead the process of identify which pixels contain clouds and which do not.

Revised.

Page 6, line 7: +/-50x50 km is a somewhat larger area than used in the Ichoku and Levy papers cited here (they use more like 25 or 27.5 km instead). It is probably worth noting that you are using different distance ranges here, rather than following the Ichoku/Levy technique exactly.

We used the same way as Ichoku et al. used, i.e., 5 pixels x 5 pixels. We have changed the presentation to +/-25x25 km. Many thanks.

Page 8, general: the terms 'lowly relevance' and 'highly relevance' are strange and not intuitive. I understand that 'lowly relevance' means data which are a bad match (larger than average error), and 'highly relevance' means data which are a good match (smaller than average error). Perhaps alternative terms could be used here, e.g. 'close to truth' or 'far from truth'?

Revised.

Section 3: I still think that splitting the results up by year is a bad idea. The sample size is too small to infer year-to-year changes in performance in a robust way, and splitting by year just adds complexity. Figure 1 now combines the years (which is good) but I would prefer if the text and tables did the same thing. However this is not a huge problem so up to the editor and authors if they wish to make further changes.

Although we have only three years data, we have done the comparisons for each year and for all three years together. We prefer to keep all analysis.

Section 3.3: I still think that the data volume is too low to infer the year-to-year variability of performance at these sites. I would prefer that this analysis were therefore either removed, or a clear caveat about the data volume put at the start. The analysis gives some useful information but it is important not to over-interpret results from a small selection of data points.

Have added a few lines for it.

Page 26, line 28: an unstable data server URL is not appropriate in a journal for a reference about what Angstrom exponent is. This should be replaced with a proper reference. Eck et al (1999), doi:10.1029/1999JD900923 is one paper commonly used as a citation about this. Or, the authors could go back to Angstrom's original papers.

Revised. Many thanks.

[revised manuscript text omitted]